# Long-term relapse-free survival enabled by integrating targeted antibacteria in antitumor treatment

Yuanlin Wang[1], Yaqian Han[2], Chenhui Yang[1], Tiancheng Bai[1], Chenggang Zhang[2], Zhaotong Wang[2], Ye Sun [2] ✉, Ying Hu [3], Flemming Besenbacher [4], Chunying Chen [5] & Miao Yu [1] ✉

The role of tumor-resident intracellular microbiota (TRIM) in carcinogenesis has sparked enormous interest. Nevertheless, the impact of TRIM-targeted antibacteria on tumor inhibition and immune regulation in the tumor microenvironment (TME) remains unexplored. Herein, we report long-term relapse-free survival by coordinating antibacteria with antitumor treatment, addressing the aggravated immunosuppression and tumor overgrowth induced by TRIM using breast and prostate cancer models. Combining $Ag^+$ release with a Fenton-like reaction and photothermal conversion, simultaneous bacteria killing and multimodal antitumor therapy are enabled by a single agent. Free of immune-stimulating drugs, the agent restores antitumor immune surveillance and activates immunological responses. Secondary inoculation and distal tumor analysis confirm lasting immunological memory and systemic immune responses. A relapse-free survival of >700 days is achieved. This work unravels the crucial role of TRIM-targeted antibacteria in tumor inhibition and unlocks an unconventional route for immune regulation in TME and a complete cure for cancer.

With hundreds of trillions of microorganisms inhabited in the human body, the microbiomes and their impact on various physiologic functions, disease, aging, emotion, and cognition have long been in the spotlight[1]. In particular, tumor-resident intracellular microbiota (TRIM) has lately aroused considerable concern as a significant contributor to carcinogenesis[2–4]. TRIM can be resident in tumors via migration from surrounding tissues or penetration through the leaky vasculature in tumors upon blood circulation[4,5], where the immunosuppressed tumor microenvironment (TME) provides TRIM a refuge[5]. There is growing evidence suggesting that TRIM can decrease the efficacy of chemotherapy, enhance the toxicity of drugs to normal tissues, and promote the pro-inflammatory response and the proliferation of tumor cells[6–9]. Significant progress has been made in enhancing anticancer efficacy and immunity responses by regulating the intestinal microbiota[10,11]. Still, the impact of bacteria killing in TME on tumor inhibition remains unclear.

The common strategy to eliminate TRIM relies on a systemic dose of antibiotics. Lacking targeting[6,12], the systemic dose can induce a series of negative effects[13,14], including intestinal probiotic deficiency, disrupted homeostasis, disease susceptibility, drug resistance, etc., especially for patients with malignancy. In addition, the use of antibiotics can compromise the effectiveness of immune checkpoint inhibitions in anticancer treatment[15], and seriously reduce the efficacy of chemotherapy on melanoma and lung cancer[16].

[1]State Key Laboratory of Urban Water Resource and Environment, School of Chemistry and Chemical Engineering, Harbin Institute of Technology, Harbin 150001, China. [2]School of Instrumentation Science and Technology, Harbin Institute of Technology, Harbin 150001, China. [3]School of Life Science and Technology, Harbin Institute of Technology, Harbin 150001, China. [4]Interdisciplinary Nanoscience Center (iNANO), Aarhus University, Aarhus 8000, Denmark. [5]National Center for Nanoscience and Technology, Chinese Academy of Sciences, Beijing 100190, China. ✉e-mail: sunye@hit.edu.cn; miaoyu_che@hit.edu.cn

TRIM-targeted antibacteria is, therefore, desirable but yet to be realized.

Beyond the short-term inhibition rate, the ultimate antitumor goal is long-term relapse-free survival, which necessitates lasting immunological memory against recurrence as well as systemic immune responses addressing cancer metastasis. Various therapies developed in recent years, e.g., chemodynamic therapy (CDT) and photothermal therapy (PTT), have shown remarkable short-term efficacy; nevertheless, residual cancer or mutant cells in TME still induce a high risk of tumor recurrence[17–19]. To improve relapse-free survival, a combination of these antitumor therapies[20,21] with immune activation based on the use of immune-stimulating drugs (e.g., immune modulators, immune checkpoint blockades, antigens, etc.) has been proposed[22–24]. There is no precedent for combining TRIM-targeted antibacteria with antitumor therapy. The antitumor efficacy and how the immune system would respond to such a combination are open questions.

Herein, we report long-term relapse-free survival achieved by integrating TRIM-targeted antibacteria into antitumor therapies, addressing the immunosuppression and overgrowth of TRIM-infected carcinoma. Breast carcinoma is selected as the prototype, considering its markedly rising incidence/mortality rate as the top life-threatening cancer for women[25,26] and the known abundance/diversity of microbiota in the mammary gland[5,27]. *Escherichia coli* (*E. coli*), a Proteobacteria strain abundant in both malignant and normal breast cells[4,5], is employed to construct the TRIM model.

We demonstrate that the TRIM can upregulate the expression of immunosuppressive cytokines [interleukin-10 (IL-10), transforming growth factor-β (TGF-β)] and pro-inflammatory cytokines IL-17 while downregulating the expression of pro-inflammatory cytokines [including IL-12, tumor necrosis factor-α (TNF-α), and interferon-γ (IFN-γ)] and programmed cell death-1 (PD-1) cytokines[12,28]. TRIM also reduces the number of T cells and M1-like tumor-associated macrophages (TAMs) while increasing M2-like TAMs. Consequently, immunosuppression in the TME is reinforced, and tumor growth is accelerated. To enable simultaneous antibacteria with antitumor therapies by a single agent, we have designed the Ag₂Se shell-covered Au nanoparticles with their surface modified by folic acid (Au@Ag₂Se-FA NPs, Fig. 1). The released Ag ions and Fenton-like reaction endow powerful antibacterial, together with chemotherapy and CDT, to inhibit both the TRIM and tumor cells, promoting M2-to-M1-like TAM

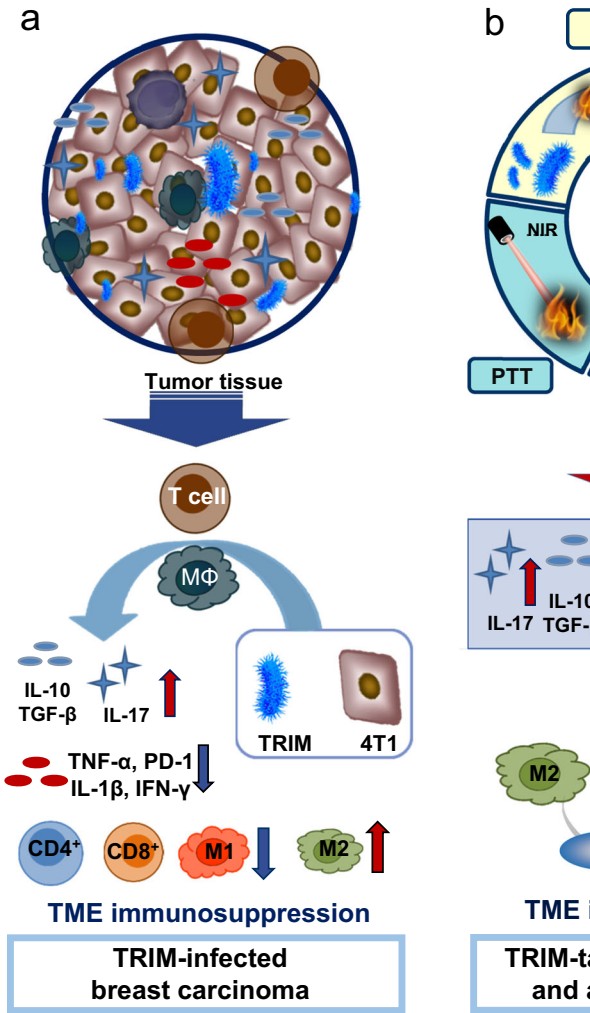

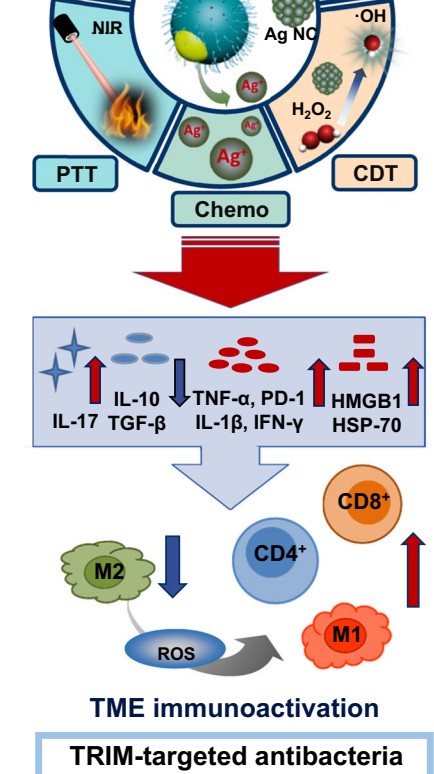

**Fig. 1 | Schematic diagram integrating TRIM-targeted antibacteria with antitumor therapies. a** The aggravated immunosuppression induced by TRIM in the TME, where the presence of TRIM upregulates the expression of IL-17, TGF-β, and IL-10 while downregulates PD-1, TNF-α, IL-1β, IL-12, and IFN-γ, reducing the number of T cells and M1-like type TAMs but increasing M2-like type TAMs. **b** Simultaneous TRIM-targeted antibacteria and multimodal antitumor therapy (CDT, PTT, and chemotherapy) enabled by a single agent, i.e., the Ag₂Se shell covered-Au NPs equipped with folic acid (Au@Ag₂Se-FA): the released Ag ions and Fenton-like reaction of Ag NCs evolved upon NIR irradiation endow antibacteria together with chemotherapy and CDT to inhibit both the TRIM and tumor cells; the high extinction coefficient and photothermal conversion efficiency of Au@Ag₂Se-FA allow significant hyperthermia to eliminate cancer cells (PTT). The combination upregulates PD-1, IL-17, TNF-α, IFN-γ, and IL-1β, downregulates IL-10 and TGF-β, releases HSP-70 in the TME, activates the immune response of T cells, and promotes the repolarization of TAMs from the M2- to M1-like phenotype.

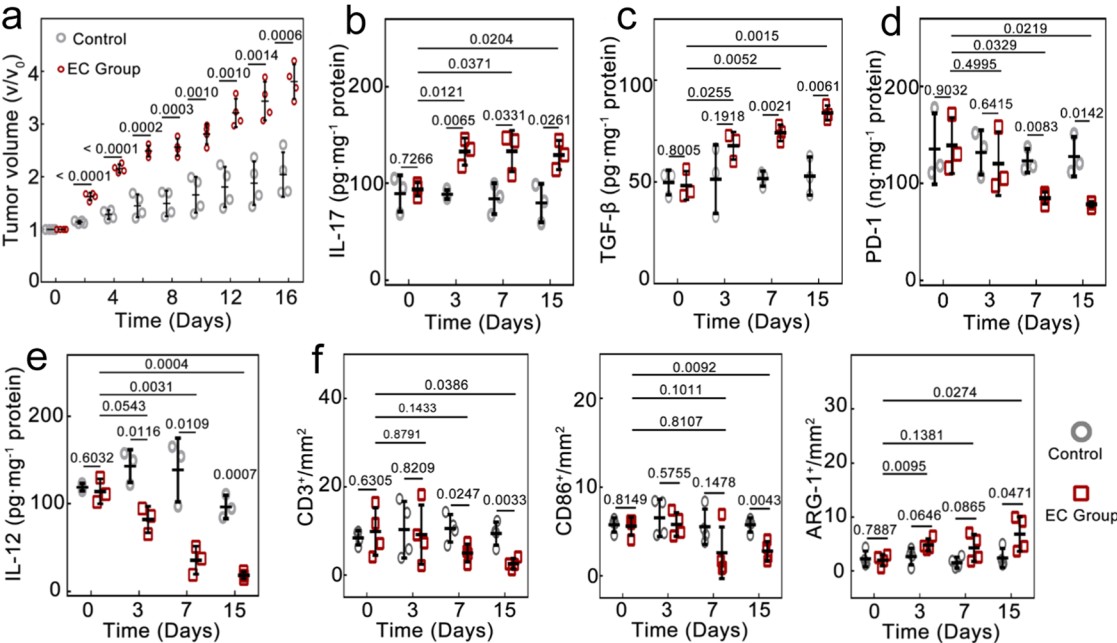

**Fig. 2 | TRIM-induced immunosuppression and overgrowth of breast tumors.**
**a** Tumor growth rate of the *E. coli*-infected group (EC Group) and the non-infected group (Control) (*n* = 4 mice per group). Compared with Control, the tumors in EC grow evidently faster, suggesting that the TRIM can promote tumor growth. The intratumoral expression of (**b**), IL-17, (**c**), TGF-β, (**d**), PD-1, and (**e**), IL-12, analyzed by enzyme-linked immunosorbent assay (ELISA, *n* = 3 mice per group), showing that the presence of TRIM can increase the expression of IL-17 and TGF-β, which are known for promoting breast tumor antiapoptotic, proliferation, and invasiveness, and downregulate PD-1 and IL-12. The variations indicate the aggravated immunosuppression and weakened responses of T cells and M1-like TAMs in the TME. **f** Immunohistochemistry analysis of T cells (CD3+), M1-like TAMs and dendritic cells (CD86+), and M2-like TAMs (ARG-1+) in the tumors, respectively (*n* = 4 mice per group). Data are presented as mean ± standard deviation. One-way ANOVA with Tukey's post hoc test. *P* < 0.05 is considered to be statistically significant.

repolarization. PTT allows high hyperthermia upon near-infrared (NIR) irradiation to eliminate cancer cells, releasing heat shock protein (HSP)-70 in the TME. Pronounced immune responses are activated. The immunological regulation of this antibacteria-combined antitumor therapy is effective in both TRIM-infected mice and those raised in a non-sterile environment. Secondary inoculation and distal tumor studies confirm the presence of long-lasting immunological memory and systemic immune responses. As a result, the breast tumor is cured, with no recurrence for at least 700 days. A second cancer model, prostate cancer, demonstrates the generality of this approach. This work unravels the importance of integrating TRIM-targeted antibacteria in antitumor treatment for tumor inhibition and immune regulation in the TME.

## Results

### Impact of TRIM on breast tumor

To construct the TRIM model, $1 \times 10^5$ CFU *E. coli* cells were intratumorally (*i.t.*) injected into 4T1 breast tumor-bearing mice (EC), with the mice *i.t.* injected with sterile phosphate buffer solution (PBS) as a control. The number of bacteria was chosen to be representative of a normal infection[6,12,29]. The injected *E. coli* were alive in the tumors and proliferated steadily over time (Supplementary Fig. 1). For instance, the bacteria density increased from $2.3 \times 10^4$ CFU g$^{-1}$ at 12 h to $5.5 \times 10^7$ CFU g$^{-1}$ at Day 7 (i.e., 7 days after injection). The tumor growth was monitored: while the tumor volume in the control group enlarged to 204.1% at Day 16, the tumors in the EC group already increased to 215.6% at Day 4 and to 380.6% at Day 16 (Fig. 2a); the mean tumor weight in the EC group was consistently higher than that in Control (Supplementary Fig. 2). The results indicated that TRIM promoted tumor growth. To explore the immunologic state of T cells and TAMs in the TME, we evaluated the expression of IL-17, PD-1, IL-12, and TGF-β, where the former two markers are key indicators for the activation of

intratumoral T cells and the latter two are crucial for evaluating M1-like and M2-like TAMs[12]. Evidently, both IL-17 (Fig. 2b) and TGF-β (Fig. 2c) of EC upexpressed, increasing to 150%, 158%, and 162% (for IL-17), and 132%, 144%, and 159% (for TGF-β) as compared to Control at Day 3, 7 and 15, respectively. The PD-1 (Fig. 2d) and IL-12 expression (Fig. 2e) of EC showed a substantial decline, reaching only 61% and 16% as those of Control at Day 15. In contrast, there were no obvious changes in the above-mentioned markers in the tumors in the control group. Furthermore, as revealed by the immunohistochemistry (IHC) analysis (Fig. 2f and Supplementary Fig. 3), the numbers of T cells (CD3+) and M2 TAMs (ARG-1+) evidently varied after *E. coli* injection: at Day 15, the number of T cells reduced to 27%, while M2 TAMs increased to 286%. In contrast, these cell numbers of the control group were barely varied. Consistently, flow cytometric analysis (Supplementary Fig. 4) demonstrated that *E. coli* injection reduced the number of M1-like TAMs (CD11b+, CD86+) to 65%. All these results indicated aggravated immunosuppression in the TME by the TRIM; the simultaneous increase of IL-17 and TGF-β suggested that the IL-17 pathway may play a crucial role in breast tumor overgrowth, similar to the case of TRIM-infected colorectal cancer reported previously[28].

### Design and characterization of Au@Ag$_2$Se-FA

Considering the known bacteria inhibition effect of Ag ions[30], the immunity and antibacteria enhancement of Se[31,32], the photothermal conversion and biocompatibility of metal selenide[33,34], and the boosted photon harvesting and reactive oxygen species (ROS) generation of the Au-metal selenide hybrid[33,34], we fabricated Au@Ag$_2$Se. To enable tumor targeting, FA was equipped on the NPs' surface (Fig. 1 and Supplementary Fig. 5), and the resultant Au@Ag$_2$Se-FA NPs were used in all studies. Transmission electron microscopy (TEM) images (Fig. 3a and b) and elemental mapping (Fig. 3c) revealed that the Au@Ag$_2$Se-FA NPs had an asymmetric core-shell structure and an

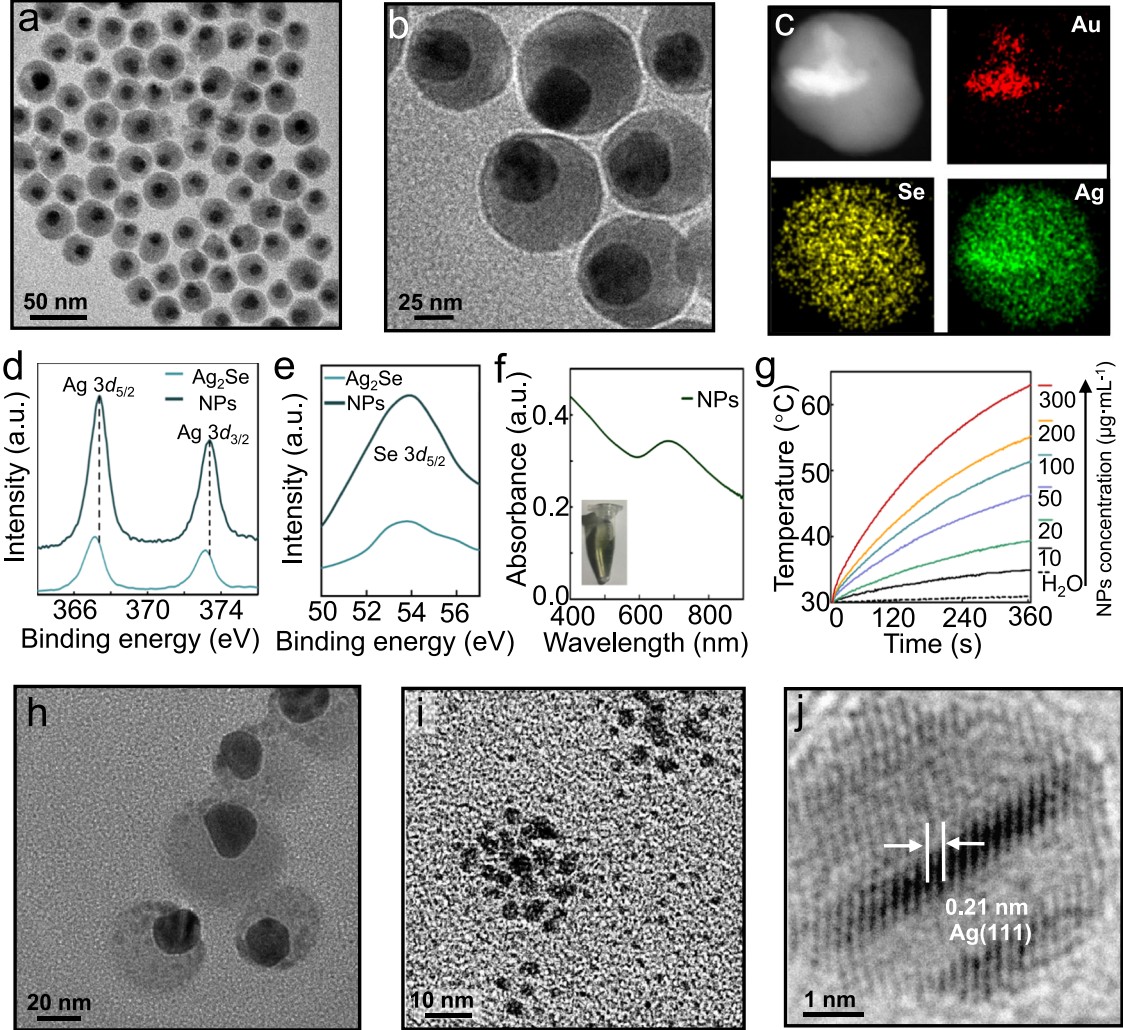

**Fig. 3 | Characterization of the Au@Ag₂Se-FA NPs. a** TEM, and (**b**), high-resolution (HR) TEM images of Au@Ag₂Se-FA NPs, showing an asymmetric core-shell structure where the dark circular core and the half-transparent shell are attributed to Au and Ag₂Se, respectively. **c** Elemental mapping of an individual Au@Ag₂Se-FA NP, where the signals of Au, Ag, and Se are in red, green, and blue, respectively. **d** Ag 3*d*, and (**e**), Se 3*d* X-ray photoelectron spectra of Au@Ag₂Se-FA NPs (NPs), with the spectra of Ag₂Se NPs as a comparison: the peaks located at 367.4 and 373.4 eV in panel d are attributed to Ag⁺, and the peak centered at 53.8 eV in panel e is attributed to Se²⁻ of Au@Ag₂Se-FA. **f** UV−Vis−NIR absorption spectrum of Au@Ag₂Se-FA dispersion (100 μg mL⁻¹), where the inset shows its digital photo. **g** Concentration-dependent temperature rise of Au@Ag2Se-FA dispersion at various concentrations (0−300 μg mL⁻¹) upon NIR irradiation (808 nm, 1.2 W cm⁻²). **h** TEM image of Au@Ag₂Se-FA after irradiation, showing dotted-like motifs formed in the Ag₂Se shell. **i** TEM image of the small NPs (<5 nm) in the supernatant of irradiated Au@Ag₂Se-FA dispersion. **j** HRTEM image of a single small NP in (**i**), revealing its crystallization and a lattice spacing (∼0.21 nm) matching that of Ag(111).

average total size of ∼35 nm, where the dark circular core and half-transparent shell were attributed to Au and Ag₂Se, respectively. Dynamic light scattering (DLS) was used to determine the hydrodynamic size in various physiological solutions (Supplementary Fig. 6), revealing a *Z*-average of 49.4 nm and a polydispersity index of 0.03. X-ray photoelectron spectroscopy (XPS) analysis (Fig. 3d and e) confirmed the form of Ag⁺ and Se²⁻ in the NPs, where the slight redshift of the Ag 3*d* peaks (from 367.2 and 373.2 eV to 367.4 and 373.4 eV)[35] suggested the coupling between Ag₂Se and Au.

The ultraviolet-visible-near-infrared (UV−Vis−NIR) spectrum of Au@Ag₂Se-FA showed strong and broad NIR absorption (Fig. 3f). The linear increase of absorbance at 808 nm with the NPs' concentration revealed good dispersibility (Supplementary Fig. 7), yielding an extinction coefficient of ∼13.64 L g⁻¹ cm⁻¹. Au@Ag₂Se-FA showed rapid temperature rise upon 808 nm irradiation (1.2 W cm⁻², Fig. 3g). For instance, at a concentration as low as 100 μg mL⁻², the dispersion temperature rose from 30.0 to 50.9 °C within 6 min, whilst deionized (DI) water (Control) scarcely increased in temperature (<1.9 °C) under

the same irradiation conditions. Photothermal conversion efficiency was measured to be ∼68.0% (Supplementary Fig. 8 and Supplementary Table 1).

Au@Ag₂Se-FA showed good storage stability in various physiological solutions at room temperature (RT) (Supplementary Fig. 6). The Ag⁺ release rate was determined by inductively coupled plasma mass spectroscopy (ICP-MS), showing barely any release (<1.5%) when keeping the Au@Ag₂Se-FA dispersions at 37 °C for 48 h (Supplementary Fig. 9). Upon NIR radiation, dotted morphology appeared in the shell of Au@Ag₂Se-FA (Fig. 3h); Ag nanocrystals (NCs) (<5 nm) were observed in the supernatant (Fig. 3i), showing typical crystallinity (Fig. 3j). Based on the XPS analysis (Supplementary Fig. 10), after irradiation, both peaks in the Ag 3*d* spectrum had a redshift of 0.2 eV and the peak in the Se 3*d* spectrum had a blueshift of 0.7 eV. The variations were attributed to the Ag vacancies caused by the formation of Ag NCs during irradiation. Although the profile of the Ag₂Se shell was maintained after irradiation, the composition was transformed to non-stoichiometric Ag₂₋ₓSe. The evolution of Ag NCs benefited Ag⁺

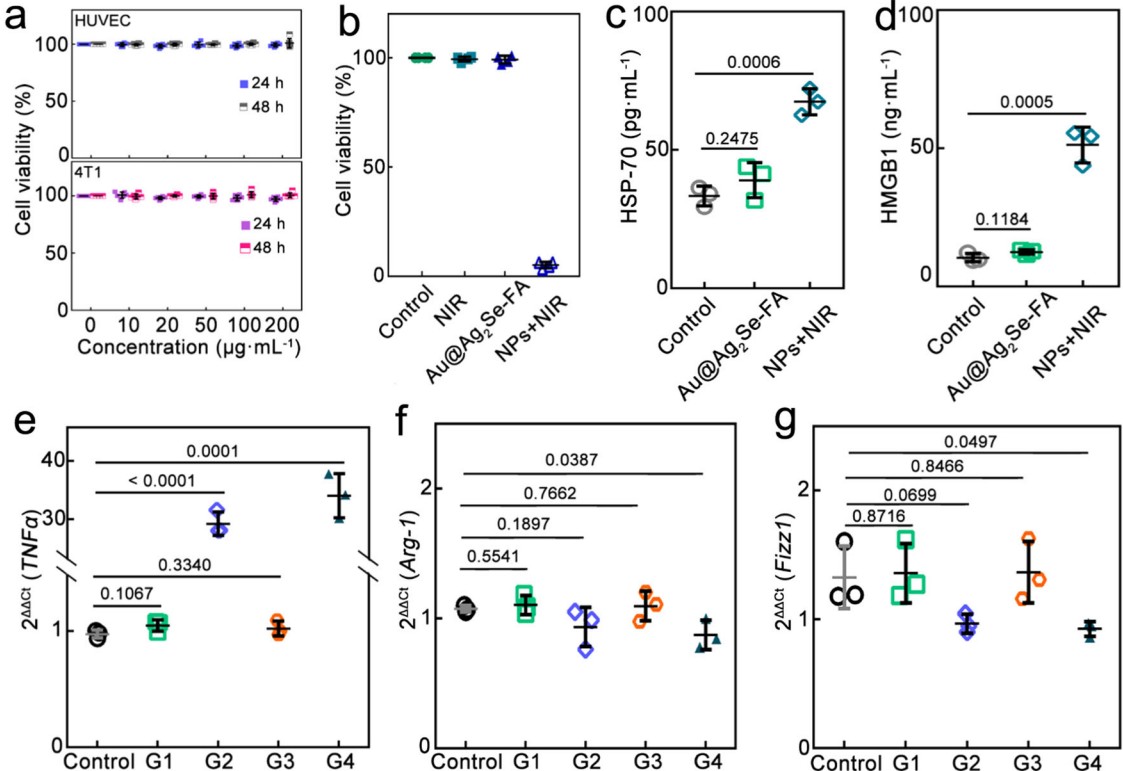

**Fig. 4 | Therapeutic efficacy and immune response induced by Au@Ag₂Se-FA in vitro. a** Cell viability of HUVEC (normal cells) and 4T1 cells (breast carcinoma cells) after incubation with Au@Ag₂Se-FA dispersions at various concentrations for 24 h and 48 h, respectively (n = 6 experimental replicates). **b** Cell viability of 4T1 cells treated by (i) NIR irradiation only (0.8 W cm⁻², 10 min) (NIR), (ii) Au@Ag₂Se-FA only (100 μg mL⁻¹) (Au@Ag₂Se-FA), and (iii) Au@Ag₂Se-FA plus (100 μg mL⁻¹) combined with NIR irradiation (0.8 W cm⁻², 10 min) (NPs + NIR), with the untreated cells as Control (n = 4 experimental replicates). **c** Analysis of HSP−70 level in 4T1 cells after treated by Au@Ag₂Se-FA only and NPs+NIR using ELISA (n = 3 experimental replicates). **d** HMGB1 level in the supernatants of 4T1 cells after various treatments (n = 3 experimental replicates), including (i) 4T1 cells co-cultured

with BMDMs for 12 h (Control), (ii) 4T1 cells cultured with Au@Ag₂Se-FA for 12 h and then co-cultured with BMDMs for another 12 h (Au@Ag₂Se-FA), and (iii) 4T1 cells treated by NPs + NIR and then co-cultured with BMDMs for 12 h (NPs+NIR). The relative expression of (**e**) TNFα, (**f**), Arg-1, and (**g**), Fizz1 from M0 macrophages treated with (i) Au@Ag₂Se-FA (100 μg mL⁻¹) (G1), (ii) PI-Au@Ag₂Se-FA (100 μg mL⁻¹) (G2), (iii) NIR irradiation only (0.8 W cm⁻², 10 min) (G3), and (iv) Au@Ag₂Se-FA plus NIR irradiation (G4) (n = 3 experimental replicates). Data are presented as mean ± standard deviation. One-way ANOVA with Tukey's post hoc test. P < 0.05 is considered to be statistically significant. All of these in vitro experiments have been conducted at least three times, yielding well consistent results.

release: Ag⁺ release of >24.1% was detected after only 10 min irradiation (Supplementary Fig. 9).

## Therapeutic efficacy in vitro
Evaluated by the Cell Counting Kit-8 (CCK-8) assay, cell viability of both human umbilical vein endothelial cells (HUVEC, normal cells) and 4T1 cells (breast carcinoma cells) was maintained >97.2% after incubation with Au@Ag₂Se-FA at various concentrations for 24/48 h, indicating the rather low/no cytotoxicity of Au@Ag₂Se-FA (Fig. 4a). Hemolytic analysis showed that Au@Ag₂Se-FA was blood compatible (Supplementary Fig. 11). To explore the anticancer efficacy in vitro, 4T1 cells were treated with (i) NIR irradiation alone (0.8 W cm⁻², 10 min), (ii) Au@Ag₂Se-FA (100 μg mL⁻¹) alone, and (iii) Au@Ag₂Se-FA combined with NIR irradiation (NPs + NIR), respectively. Unlike the former two, which showed barely any inhibition effect, the third killed 94.8% of 4T1 cells (Fig. 4b). The therapeutic effect was attributed not only to the hyperthermia induced by PTT (Supplementary Fig. 12) but also to the hydroxyl radical (a particularly aggressive ROS species inducing tumor inhibition) generated in CDT and chemotherapy. To exclude the effects of hyperthermia and incident photons, we investigated the case in which the Au@Ag₂Se-FA dispersion was pre-irradiated (PI-Au@Ag₂Se-FA) and no subsequent irradiation was applied. Similar to NPs + NIR, a strong green signal was visualized from the 2,7-dichlorodihydrofluorescein diacetate-stained 4T1 cells after incubation with PI-Au@Ag₂Se-FA, indicating the remarkable

generation of ROS (Supplementary Fig. 13). In contrast, no ROS signal was observed in the cells treated with pristine Au@Ag₂Se-FA without irradiation. Moreover, the increased G1 population and decreased G2 population of 4T1 cells revealed disturbed cell mitosis, hence promoting apoptosis by PI-Au@Ag₂Se-FA (Supplementary Figs. 14 and 15). In addition, efficient bacterial ablation was demonstrated (Supplementary Fig. 16): NPs+NIR killed 99.5% of E. coil at a low concentration of 100 μg mL⁻¹ after 10 min irradiation (without subsequent incubation); PI-Au@Ag₂Se-FA delivered an antibacteria rate of 99.5% at 500 μg mL⁻¹ after 24 h incubation, which was comparable to that of ceftriaxone (a powerful broad-spectrum antibiotic in clinic, with a rate of 99.7% at 500 μg mL⁻¹).

## Immune response in vitro
Treated by NPs+NIR, the expression of HSP-70 in the 4T1 cells increased to 202% compared to untreated cells (Control, Fig. 4c and d). The HSP-70 upregulation may be attributed to the hyperthermia in PTT, benefiting the expression of antigens and multiple cytotoxic lymphocytes in tumors[36]. Next, we cultured 4T1 cells with bone marrow-derived macrophages (BMDMs) (Fig. 4d) to evaluate the release of high mobility group box1 (HMGB1) upon macrophage phagocytic clearance[22]. Au@Ag₂Se-FA alone could not regulate HMGB1. The high HMGB1 level of NPs + NIR (487% as that of Control) indicated secondary necrosis and enhanced immune responses[37,38] even upon macrophage clearance. Human leukemia monocyte THP-1 cells were

treated with 12-O-tetradecanoylphorbol-13-acetate to get M0 macrophages[39]. After being treated with (i) Au@Ag$_2$Se-FA alone (G1), (ii) PI-Au@Ag$_2$Se-FA alone (G2), (iii) NIR irradiation only (G3), and (iv) Au@Ag$_2$Se-FA plus NIR irradiation (G4), all groups showed high cell viability (>95.5%), suggesting rather low/no cytotoxicity to M0 macrophages (Supplementary Fig. 17). Reactive nitrogen species (RNS) and ROS were largely promoted in M0 macrophages of G2 and G4, with RNS being 242% and 251% and ROS being 178% and 195% as that of Control, respectively (Supplementary Fig. 18). High ROS and RNS levels can act as second messengers in macrophages, stimulating pro-inflammatory signaling cascades and promoting M1-like transformation[40]. This was supported by the upregulated expression of IL-12 in G2 and G4 (Supplementary Fig. 19). The relative expression of *TNFα*, *Arg-1*, and *Fizz1* was evaluated using quantitative real-time polymerase chain reaction (qRT-PCR): PI-Au@Ag$_2$Se-FA upregulated *TNF-α* to 27.86-folds (Fig. 4e) and downregulated *Arg-1*, *Fizz-1* to 87% and 73% (Fig. 4f and g) as that of Control, respectively, indicating the M0/M2-to-M1-like transformation of TAMs.

## Biodistribution and excretion of the NPs
The biodistribution of Au@Ag$_2$Se-FA was assessed by measuring the content of Au ions in the major organs of the tumor-bearing BALB/c mice at 12 h after intravascular (*i.v.*) injection. A high level of ~17.8% ID g$^{-1}$ was detected from the tumor area, indicating efficient uptake of Au@Ag$_2$Se-FA by the tumor (Supplementary Fig. 20). The agent's tumor-targeting accumulation was also verified by the substantial tumor-selective photothermal effect after *i.v.* injection of Au@Ag$_2$Se-FA (Supplementary Fig. 21), where the surface modification of FA promoted the tumor-targeting effect. The levels in mononuclear phagocyte system organs, i.e., liver, spleen, and kidney, were 31.6% ID g$^{-1}$, 10.1% ID g$^{-1}$, and 7.5% ID g$^{-1}$, respectively. The urinary excretion of Au@Ag$_2$Se-FA was also investigated (Supplementary Fig. 22): ~12% and 10% of Au and Ag were excreted in urine five days after injection, and ~49% of Se was excreted in urine three days after injection. The high concentration of Se, Ag, and Au ions detected in urine suggested that Au@Ag$_2$Se-FA NPs could be metabolized via the kidney and excreted out of the body in the urine. Combined with the biodistribution of Au@Ag$_2$Se-FA in the mononuclear phagocyte system organs, the vital role of nephritic and hepatic clearance in the excretion of degradation products was confirmed.

## Therapeutic efficacy in vivo
Tumor-bearing BALB/c mice were divided into six groups, three of which were *i.t.* injected with *E. coli* (T2, T4, and T6). All groups were then treated by (i) PBS with NIR irradiation (T1 and T2), (ii) Au@Ag$_2$Se-FA without irradiation (T3 and T4), and (iii) Au@Ag$_2$Se-FA plus irradiation (T5 and T6). In all cases, *i.v.* injection of Au@Ag$_2$Se-FA (or PBS) was applied. 12 h after injection, irradiation (0.36 W cm$^{-2}$, 10 min) was carried out, and the tumor area temperature was monitored by infrared thermal (IRT) imaging (Fig. 5a). After the treatments, the tumors' volume was measured (Fig. 5b): the tumors treated by NIR irradiation or Au@Ag$_2$Se-FA alone (T1–4) grew swiftly, especially those infected with the TRIM (T2, T4). Significant tumor inhibition was attained by NPs+NIR (T5, T6) (Fig. 5b, Supplementary Figs. 23 and 24), delivering a high inhibition rate of ~100% at Day 16. The residual *E. coil* level in T6 tumor tissues was examined (Supplementary Fig. 25), confirming the potent antibacteria of NPs+NIR. The mice survival rate of T5–6 showed no difference (Fig. 5c); remarkably, neither mouse death nor tumor recurrence was observed >700 days after the treatment, realizing a complete cure.

## Immune response in vivo
The T2 group showed a stronger immunosuppression effect than T1: the helper T cells (CD3$^+$, CD4$^+$), cytotoxic T cells (CD3$^+$, CD8$^+$), and M1-like TAMs (CD86$^+$, CD11b$^+$) in T2 decreased to 59%, 27%, and 60%,

respectively, while M2 TAMs (CD206$^+$, F4/80$^+$) increased to 141% as that of T1 (Supplementary Fig. 26). The variation between T1 and T2 further confirmed the aggravated antitumor immunosuppression effect of TRIM in breast cancer. When treated by NPs + NIR, even upon the immunosuppression reinforced by the TRIM, the immune response in the T6 group was still activated effectively (Fig. 5d): the expression of TNF-α, IL-1β, IL-17, IFN-γ, and PD-1 increased to 391%, 219%, 119%, 312%, and 193%, while IL-10 and TGF-β decreased to 23% and 32%, compared with that of T2, respectively. The decreased TGF-β but increased IL-17, TNF-α, IL-1β, and IFN-γ suggested promoted cell apoptosis in the TME[41]. Importantly, the level of pro-inflammatory cytokines including TNF, IL-6, IFN-γ, and IL-1β in serum increased quickly after the NPs + NIR treatment, reaching the peak at 72 h (Fig. 5e); afterwards, the level of these markers gradually declined, restoring the baseline level at Day 7. These results indicated that the systemic immunostimulatory effect induced by the treatment can be confined within a reasonable period, which is preferred in clinics.

## Long-lasting immunological memory
To demonstrate the long-term durability of the immune responses, 45 days after the treatment, mice of T5 and T6 (*n* = 9) that had survived from the first inoculation of 4T1 tumors were re-challenged via subcutaneous administration of 4T1 tumor cells (1.0 × 10$^6$) at the previously tumor-bearing area without any additional treatments (R-T5 and R-T6 groups). 10 days after the re-challenge, the mice's blood and spleens were evaluated by flow cytometric analysis (Fig. 5f, Supplementary Figs. 27–29). Excitingly, all these mice still exhibited significant immune responses. The number of NK cells (CD3$^-$, CD49b$^+$)[42] in R-T5 and R-T6 grew to 455% and 481%, the γδ T cells (CD3$^+$, TCRβ$^-$)[43] increased to 363% and 389%, and the memory T cells (CD8$^+$, CD44$^{high}$, CD62L$^{low}$)[22] increased to 601% and 606%, as compared to the untreated tumor-bearing mice (Control). Both R-T5 and R-T6 showed no sign of tumors 21 days after the re-challenge (Supplementary Fig. 30). The results emphasized the long-lasting immunological memory against 4T1 malignancy by NPs+NIR treatment. Such memory may have played a crucial role in the complete cure.

## Effect on the distal tumor
To explore the systemic immunological effects on the distal tumor (mimicking tumor migration), we established a dual-tumor model and only treated the primary tumor with *i.t.* injection of Au@Ag$_2$Se-FA NPs plus NIR, leaving the distal tumor untreated (Supplementary Fig. 31). Remarkably, effective systemic antitumor immune responses were demonstrated in the distal tumor: nine days post-treatment, the expression of TNF-α, IL-6, IFN-γ, PD-1, and IL-17 in the distal tumor increased to 205%, 245%, 192%, 514%, and 205%, while IL-10 and TGF-β decreased to 41% and 34%, respectively, compared with the distal tumor of the control group (the primary tumor was *i.t.* injected with PBS and then irradiated). Twenty-one days post-treatment, the average distal tumor weight was only 30.2% that of Control, while the primary tumor was completely eliminated, showing no recurrence. These findings suggest that such antibacteria-combined antitumor therapy can trigger pronounced systemic antitumor immune responses, potentially addressing malignant tumor migration.

## Biosafety in vivo
The agent content in the intestine was detected to be 6.7% ID g$^{-1}$ in terms of Au ions, corresponding to 56 μg mL$^{-1}$ of Au@Ag$_2$Se-FA. Even at a higher concentration (100 μg mL$^{-1}$), Au@Ag$_2$Se-FA had no impact on the intestinal probiotics *Lactobacillus reuteri* and *Bifidobacterium longum* (Supplementary Fig. 32). The effects of various therapies on immune cytokines in serum and inguinal lymph nodes of healthy mice were also studied (Supplementary Fig. 33): Au@Ag$_2$Se-FA or the combination of Au@Ag$_2$Se-FA and irradiation could not upregulate or downregulate the key immune cytokines (IL-6, IFN-γ, IL-17, and IL-10)

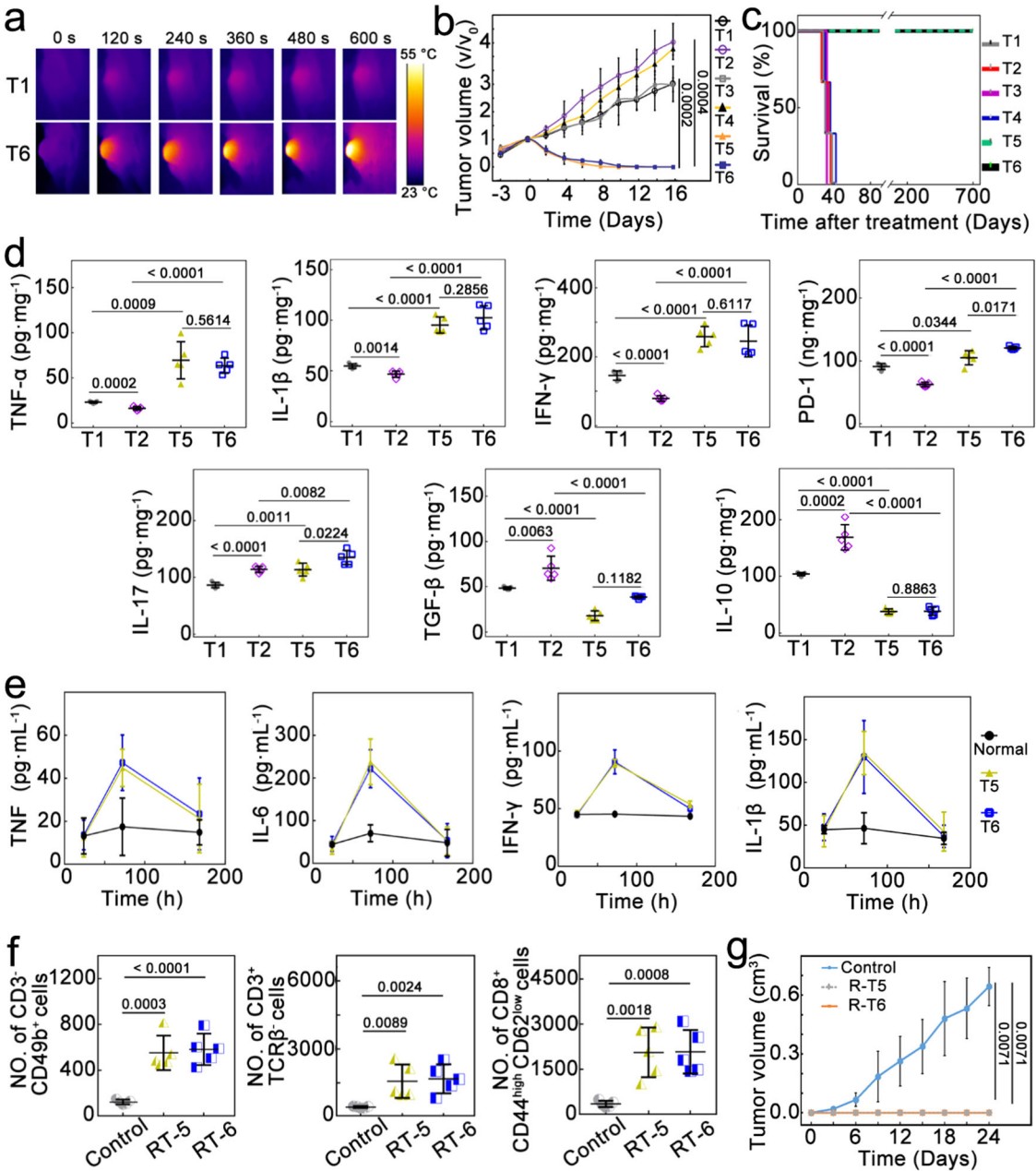

**Fig. 5 | Therapeutic effect of Au@Ag₂Se-FA in vivo. a** In vivo IRT images of 4T1 tumor-bearing mice in the T1 and T6 groups upon 10 min of NIR irradiation. **b** Tumor growth rate of different groups ($n = 5$ mice per group), three of which were *i.t.* injected with *E. coli* (T2, T4, and T6). All groups were then treated, respectively, by (**i**) *i.v.* injection of PBS plus NIR irradiation (T1 and T2), (**ii**) *i.v.* injection of Au@Ag₂Se-FA only (T3 and T4), and (**iii**) *i.v.* injection of Au@Ag₂Se-FA plus irradiation (T5 and T6). **c** Survival curves after the various treatments ($n = 5$ mice per group), where all mice in T5 and T6 survive over 700 days without tumor reoccurrence. **d** Intratumoral expression level of TNF-α, IL-1β, IL-17, IFN-γ, PD-1, IL-10, and TGF-β upon the various treatments ($n = 5$ mice per group). **e** Serum cytokine concentration of TNF, IL-6, IFN-γ, and IL-1β at the selected time points for T5 and T6

($n = 3$ mice per group), all quickly increasing with time, reaching the peak at 72 h, then declining to the baseline level as that of healthy mice without any treatment (Normal) at Day 7. **f** Flow cytometric analysis for the number of NK cells (CD3⁻, CD49b⁺), γδ T cells (CD3⁺, TCRβ⁻), and memory T cells (CD8⁺, CD44^high, CD62L^low) per 100,000 cells in blood or spleen of the R-T5 and R-T6 groups, with the results from the untreated 4T1 tumor-bearing mice as Control ($n = 5$ mice per group). **g** Tumor growth curves after subcutaneous re-challenge ($n = 4$ mice per group). Data are presented as mean ± standard deviation. One-way ANOVA with Tukey's post hoc test (**d**, **f**) or two-tailed Student's *t*-test (**b**, **g**). $P < 0.05$ is considered to be statistically significant.

of healthy mice. In contrast, *E. coli* injection increased the expression of IL-10 and IL-17 to 108% and 112% in serum, and 130% and 125% in inguinal lymph nodes, respectively, as that of Control. The behavior and weight of all mice after the treatments were monitored, showing no noticeable difference from healthy mice (Supplementary Fig. 34). The hematoxylin and eosin analysis and serum biochemistry assay further confirmed no/low in vivo toxicity of the treatment (Supplementary Figs. 35 and 36).

## Effects on mice raised in a non-sterile environment

To illustrate the role of antibacteria in a more general case, 4T1 breast tumor-bearing mice were raised in a non-sterile environment with a total bacteria amount of 7500 CFU/m³, simulating the normal living conditions. Compared to direct bacterial injection, cultivating mice in a non-sterile environment can better imitate real-life conditions that are more relevant to clinical scenarios. External bacteria have been demonstrated in clinics and laboratories to infect the host, causing

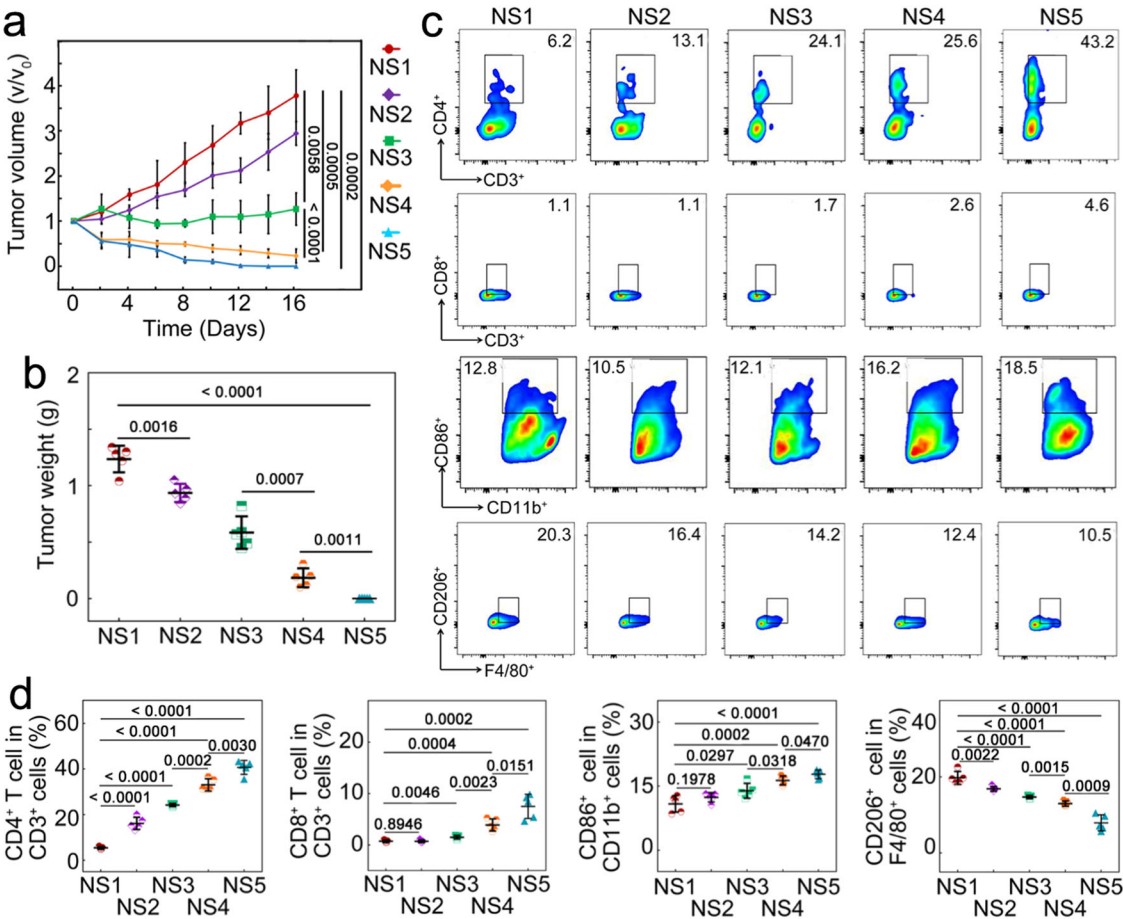

**Fig. 6 | Efficacy of Au@Ag₂Se-FA on tumor-bearing mice raised in a non-sterile environment. a** Tumor growth rate, and (**b**) tumor weight in different groups after the various treatments ($n = 5$ mice per group), including (**i**) the untreated group (NS1), the groups treated (**ii**) by ceftriaxone alone (NS2), (**iii**) by lobaplatin alone (NS3), (**iv**) by the combination of ceftriaxone and lobaplatin (NS4), and by (**v**) Au@Ag₂Se-FA plus irradiation (NS5), where NS5 delivers a tumor inhibition rate of 100% at Day 16. **c** Flow cytometric analysis of the helper (CD3⁺, CD4⁺) and cytotoxic (CD3⁺, CD8⁺) T cells, M1-like (CD86⁺, CD11b⁺), and M2-like (CD206⁺, F4/80⁺) TAMs, and (**d**) quantifications of these cells in the TME after the various treatments ($n = 5$ mice per group), where NS5 delivers the strongest immune responses of T cells and TAMs. Data are presented as mean ± standard deviation. One-way ANOVA with Tukey's post hoc test (**b**, **d**) or two-tailed Student's $t$-test (**a**). $P < 0.05$ is considered to be statistically significant.

accumulation and residence in tumors[5,6]. The tumor-bearing mice raised in the non-sterile environment were divided into five groups, including (i) the untreated group (NS1), (ii) the mice treated by ceftriaxone alone (NS2), (iii) by lobaplatin alone (NS3), (iv) by the combination of ceftriaxone and lobaplatin (NS4), and (v) by Au@Ag₂Se-FA plus irradiation (NS5). Lobaplatin (a commonly-used chemotherapeutic drug inhibiting cell proliferation by generating ROS[44]) showed an inhibition rate of 72.6% at 500 μg mL⁻¹ (Supplementary Fig. 37) but no antibacteria effect on *E. coli* (Supplementary Figs. 16 and 38). Ceftriaxone (a clinically powerful antibiotic) killed 99.7% of bacteria at 500 μg mL⁻¹ (Supplementary Fig. 16b) without any inhibition effect on cancer cells (Supplementary Fig. 37). Based on the level of intratumoral TNF-α, IL-1β, IFN-γ, PD-1, IL-17, TGF-β, and IL-10, NS1 demonstrated aggravated immunosuppression and tumor overgrowth (125% at Day 16) as compared with Control (tumor-bearing mice raised in the sterile environment, T1 in Fig. 5d) (Supplementary Figs. 39 and 40), suggesting that even the non-sterile environment can elevate the challenge of immune regulation in the TME. Tumors in NS2 showed a lower growth rate than in NS1 (with a tumor weight of 75.7% as that in NS1 on Day 16) (Fig. 6a and b). Combining the increased TNF-α level and help T cells and decreased TGF-β level and M2-TAMs of NS2 as compared with NS1 (Fig. 6c and d, Supplementary Fig. 39), these findings suggest that the antibiotic alone can already alleviate immunosuppression and retard tumor growth to some extent. NS3–5 showed an evident

increase of TNF-α, IL-1β, IFN-γ, and PD-1 and a decrease of TGF-β and IL-10. The efficacy of NS5 was most significant, where TNF-α, IL-1β, IFN-γ, PD-1, and IL-17 increased to 312%, 200%, 162%, 126%, and 140% while TGF-β and IL-10 decreased to 33% and 24% relative to the case of NS1. NS4 (the combination of antibiotic and chemotherapeutic drugs) delivered a largely promoted regulation than NS3 (lobaplatin alone). Reasonably, lacking the photothermal conversion capability, NS4 did not show any regulation on HSP-70 and HMGB1 expression as NS5 did (Supplementary Fig. 41). Moreover, NS4 and NS5 showed strong immune responses of T cells and TAMs (Fig. 6c and d); especially for NS5, the fraction of helper and cytotoxic T cells increased to 632% and 1043%, the level of M2 TAMs decreased to 39%, whilst the level of M1-like TAMs increased to 156% as that of NS1. The NS2, NS4, and NS5 groups showed strong antibacteria effect (Supplementary Fig. 38). NS3 showed even slower tumor growth than NS2, with a tumor volume of 126.7% on Day 16 compared to that on Day 0. NS4 delivered an inhibition rate of 77.4% 16 days post treatment. NS5 provided a tumor inhibition rate of 100%. All these results emphasized the importance of combining antibacteria with antitumor therapy for immune activation, not limited to TRIM-infected tumors.

## Therapeutic efficacy on prostate cancer model
To illustrate the generality of TRIM-induced immunosuppression in the TME and the efficacy of the TRIM-targeting antibacteria-

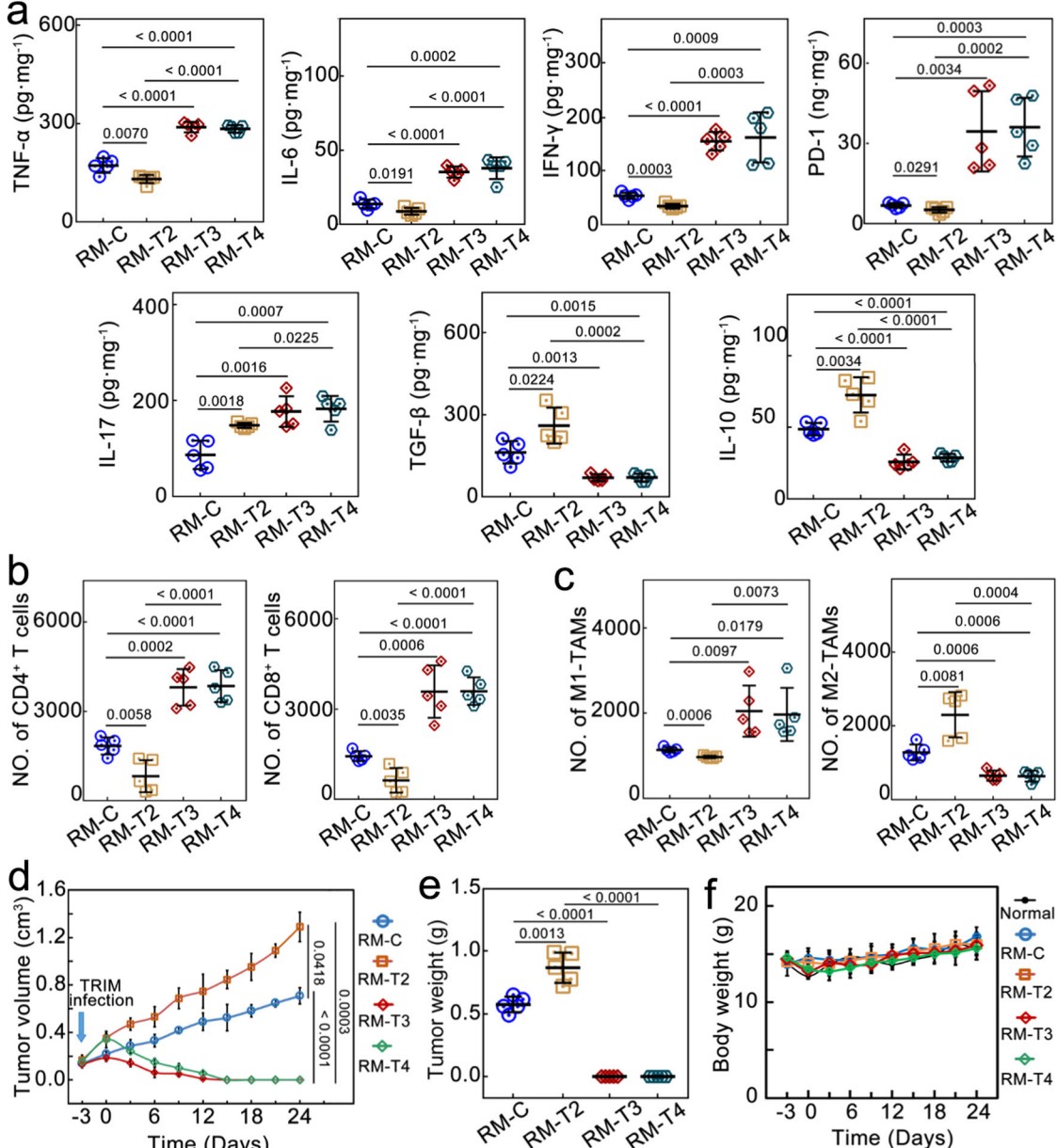

**Fig. 7 | In vivo therapeutic effect on the RM-1 cancer model. a** Intratumoral expression level of TNF-α, IL-6, IL-17, IFN-γ, PD-1, IL-10, and TGF-β upon the various treatments (*n* = 5 mice per group). **b** Flow cytometric analysis for the numbers of the helper (CD3⁺, CD4⁺) and cytotoxic (CD3⁺, CD8⁺) T cells, and (**c**) M1-like (CD86⁺, CD11b⁺) and M2-like (CD206⁺, F4/80⁺) TAMs per 100,000 cells from the tumors three days after the treatments (*n* = 5 mice per group). **d** Average tumor growth curves, (**e**) mean tumor weight, and (**f**), body weight of mice in different groups (*n* = 5 mice per group). Data are presented as mean ± standard deviation. One-way ANOVA with Tukey's post hoc test (**a**–**c**, **e**) or two-tailed Student's *t*-test (**d**). *P* < 0.05 is considered to be statistically significant.

antitumor approach, another cancer model, i.e., the prostate cancer model (RM-1), was investigated. The RM-1 tumor-bearing BALB/c mice were divided into four groups, with two groups *i.t.* injected with *E. coli* (RM-T2 and RM-T4). RM-T2 received no additional treatment. Three days later, NIR irradiation was applied to RM-T3 and RM-T4 12 h after *i.v.* injection of Au@Ag₂Se-FA. RM-C was the untreated control group. TRIM, like breast cancer, exacerbated immunosuppression in prostate cancer: six days after TRIM infection, intratumoral TNF-α, IL-6, IFN-γ and PD-1 expression in RM-T2 group decreased to 76%, 64%, 64%, and 76%, whereas IL-17, TGF-β, and IL-10 increased to 171%, 160% and 149%, respectively, compared to the control group (RM-C) (Fig. 7a). Meanwhile, helper T cells, cytotoxic T cells and M1-like TAMs in RM-T2 fell to 45%, 44%, and 85%, respectively, whilst M2 TAMs grew to 180% as compared to RM-C

(Fig. 7b, c, Supplementary Figs. 42 and 43). Tumors in RM-T2 developed faster than tumors in RM-C. NPs+NIR can effectively reverse the TRIM-induced immunosuppressive effect: the expression of TNF-α, IL-6, IFN-γ, PD-1, and IL-17 in RM-T3 (RM-T4) increased to 168% (165%), 258% (277%), 289% (302%), 511% (534%), and 204 % (211%) compared with RM-C, respectively; TGF-β and IL-10 decreased to 43% (43%), and 53% (59%), respectively; the helper and cytotoxic T cells and M1-like TAMs increased to 207% (209%), 250% (251%), and 180% (173%), respectively; M2 TAMs decreased to 51% (50%). Consistently, significant tumor inhibition was attained by NPs+NIR (RM-T3 and RM-T4) (Fig. 7d–f, Supplementary Fig. 44), delivering a high inhibition ratio of ~100%. No prostate tumor recurrence occurred one month after the therapy, demonstrating the universality and long-term efficacy of our therapeutic approach.

## Discussion

As demonstrated above, long-term relapse-free survival has been achieved by integrating antibacteria with multimodal antitumor therapy (PTT, CDT, chemotherapy) mediated by Au@Ag$_2$Se-FA. The NIR-responsive photothermal effect is attributed to the strong optical absorption and high photothermal conversion efficiency of the hybrid of Ag$_2$Se shell and Au core. CDT is based on the effective generation of hydroxyl radicals by the Fenton-like reaction of Ag NCs[45] (Ag + H$_2$O$_2$ + H$^+$ → Ag$^+$ + ·OH + H$_2$O), where the catalytic effect of Au core[33] and the elevated temperature upon irradiation further promote the reaction. The released Ag ions are known as a chemotherapeutic drug[46] and also contribute to antibacteria, which could bind to the sulfhydryl group of metabolic enzymes inducing enzyme deactivation, form complexes with the cell wall peptide sugar to disrupt electron/oxygen transport, and cause structural denaturation of DNA thereby inhibiting DNA replication[30]. The intense redox by hydroxyl radicals of the Ag NCs and hyperthermia upon NIR irradiation further strengthen the antibacteria effect. By the antibacteria-antitumor integration, even upon TRIM-worsen immunosuppression, the responses of both T cells and M1-like TAMs can be effectively activated (Supplementary Figs. 16, 18 and 19; Fig. 4c–g): the enhanced RNS, ROS, and IL-12 level in TAMs by the Ag ions/NCs promote M2-to-M1-like repolarization; the hyperthermia upon NIR irradiation triggers the secondary necrosis and release of HSP-70 in the TME, further benefiting the T cells response and promoting M1-like TAMs.

To deconvolute the role of antibacteria, we used Au@Cu$_{2-x}$Se NPs as comparison, as they had excellent PTT and CDT effect[33] and similar morphology as that of Au@Ag$_2$Se-FA. Despite the significant hyperthermia, Au@Cu$_{2-x}$Se induced negligible antibacterial effect on *E. coli* in vitro/vivo (Supplementary Figs. 16 and 25). The strong antibacterial effect of NPs + NIR was primarily attributed to the antibacterial effect of Ag ions rather than hyperthermia. We then compared the immune response activated by Au@Ag$_2$Se-FA to that by Au@Cu$_{2-x}$Se upon NIR irradiation (Supplementary Fig. 45): although T7 also delivered largely upregulated expression of intratumoral TNF-α (207%), IL-1β (140%), IFN-γ (215%) and PD-1 (162%) and downregulated expression of IL-10 (46%) and TGF-β (55%) as that of Control (T2), the regulation extent of these markers in T7 was much weaker than that in T6, suggesting a more efficient restoration of immune surveillance by Au@Ag$_2$Se-FA; moreover, the helper T cells (CD3$^+$, CD4$^+$), cytotoxic T cells (CD3$^+$, CD8$^+$) and M1-like TAMs (CD86$^+$, CD11b$^+$) in T6 increased to 192%, 282% and 252%, whilst M2 TAMs (CD206$^+$, F4/80$^+$) fell to 58% (Supplementary Fig. 26). For the tumor inhibition efficacy, Au@Cu$_{2-x}$Se plus NIR irradiation produced a tumor inhibition rate of 83.8% on the TRIM-infected tumors (T7) at 12 days post treatment, which was lower than that on the non-infected tumor (95.7%[33]). NPs + NIR showed close inhibition rate for the tumors with/without TRIM infection (100% *vs.* 98.9%) (Fig. 5b). Moreover, evident tumor recurrence can already be observed within 16 days after the Au@Cu$_{2-x}$Se + NIR treatment. Within 58 days post treatment, every mouse in T7 passed away (Supplementary Fig. 46). In stark contrast, T6 obtained a relapse-free survival of >700 days (Fig. 5c). All these results further confirmed that combination with antibacteria played a crucial role in the immune regulation in TME.

Furthermore, considering the high-Z and high X-ray attenuation coefficient of Au[33] and the X-ray absorption-induced photoelectric or Auger electron generation by Ag NCs[47], Au@Ag$_2$Se-FA could also be a potential candidate for radiation treatment and computed tomography imaging.

In summary, we reveal that the TRIM can reduce the number of T cells and M1-like TAMs, increase the number of M2-TAMs, upregulate anti-inflammatory cytokines, and downregulate pro-inflammatory cytokines, thus worsening immunosuppression in the TME of both *E. coli*-injected mice and mice raised in a non-sterile environment. Integrating TRIM-targeted antibacteria into antitumor treatment with a single agent results in an efficient solution addressing immunosuppression and tumor overgrowth. Pronounced T cell immune response and M2-to-M1-like TAM repolarization are activated, with the systemic immunostimulatory effect confined within a reasonable period. The treatment can trigger long-term immunological memory against tumor relapse and strong systemic antitumor immune responses dealing with malignant tumor migration. A relapse-free survival of >700 days is realized. Both breast and prostate carcinoma reveal the treatment's effectiveness. This work opens up a route for efficient immune activation in the TME without the use of immune-stimulating drugs and correlates TRIM-targeting antibacteria with antitumor treatments, emphasizing the crucial role of antibacteria in immune regulation and tumor inhibition. The correlation between TRIM-targeting antibacteria and immune regulation is more profound and complicated than what has been presented in this work, deserving further investigation. Although immunotherapies have shown considerable clinical promise in specific types of malignancies, they are far less effective in many other cases. The findings in this work may provide deeper insight into the ineffectiveness of immunotherapies and inspire breakthroughs for radical antitumor treatments in clinics.

## Methods

### Ethical regulations

All animal experiments were carried out under protocols approved by the Institutional Animal Care and Use Committee of Harbin Institute of Technology. The approval number is IACUC-2019021. The maximum tumor size/burden used in the present experiments was <1800 mm$^3$, which did not exceed the committee's maximal limit (2000 mm$^3$).

### Cell lines

4T1, RM-1, HUVEC, and Tohoku Hospital Pediatrics 1 (THP-1) cells (human acute monocytic leukemia cell line) were purchased from the National Collection of Authenticated Cell Cultures (Shanghai, China). These cell lines were authenticated by their morphology and short tandem repeat genotyping, and tested for mycoplasma. *E. coli* (ATCC8739) was purchased from Institute of Applied Microbiology, Heilongjiang Provincial Academy of Sciences (Harbin, China). *L. reuteri* (337178) and *B. longum* (185971) were purchased from BeNa Culture Collection (Xin yang, China).

### Materials

Chlorauric acid (HAuCl$_4$·4H$_2$O, ≥48%), silver nitrate (AgNO$_3$, ≥99.8%), sodium citrate (C$_6$H$_5$Na$_3$O$_7$·2H$_2$O, ≥99.0%), polyvinylpyrrolidone (PVP, Mw ≈ 55,000), and ascorbic acid (AA) were purchased from Aladdin (China). Folic acid-polyethylene glycol-thiol (FA-PEG-SH, Mw ≈ 2000 Da, ≥98%) was ordered from Sigma-Aldrich (China). Sodium selenite (Na$_2$SeO$_3$, ≥97.0%) was purchased from Shenyang Huadong Chemical Reagent Technologies Co. Ltd. CCK-8 was purchased from Dojindo Laboratories. All chemicals were used as received. DI water with resistance larger than 18.2 MΩ cm was used in all experiments. RPM1 Medium 1640 basic (1X), Dulbecco's modified Eagle's medium (1X), PBS (pH = 7.4, 1X), and fetal bovine serum (FBS) were purchased from Thermo Fisher Biochemical Products (Beijing) Co., Ltd. All antibodies were purchased from BD Bioscience and used following the manufacturer's instructions.

### Synthesis of Au@Ag$_2$Se NPs

To synthesize Au NPs, C$_6$H$_5$Na$_3$O$_7$·2H$_2$O (1%, 1 mL) and HAuCl$_4$·4H$_2$O (2%, 200 μL) were injected into boiling DI water (50 mL). The mixture was heated to 120 °C and reacted upon stirring for 30 min, then cooled to RT. Citrate-stabilized Au NPs were collected by centrifugation, then dispersed in PVP aqueous solution (20 μg mL$^{-1}$, 50 mL) at RT upon vigorous stirring for 1 h. Next, Na$_2$SeO$_3$ (2%, 100 μL), AA (1.4%, 1 mL), AgNO$_3$ (2%, 100 μL), and a second dose of AA (1.4%, 1 mL) were added to the solution one by one. The solution color varied from claret to forest green. The product was then centrifuged and washed three

times with DI water. The as-prepared NPs (3 mg) were dispersed in FA-PEG-SH solution (100 μg mL$^{-1}$, 50 mL) and kept at 4 °C upon vigorous stirring for 24 h to obtain the coated NPs (Au@Ag$_2$Se-FA). The NPs were finally centrifuged and washed three times with DI water for further experiments.

## Instruments

The morphology and size of the NPs were characterized by field-emission TEM (JEM-2100). The elemental mapping of the samples was carried out using energy-dispersive spectroscopy. Optical absorption properties were analyzed by a UV−Vis−NIR spectrophotometer (Evolution 300, Thermo Scientific). Sample composition and chemical structure were analyzed by XPS (AXIS ULTRA DLD, Kratos) with an Al *Kα* source. The concentration and release of Ag ions were evaluated by ICP-MS (Optima 8300, PerkinElmer). The morphology of red blood cells (RBCs) and *E. coli* cells were imaged by scanning electron microscopy (SEM, JME-7500F).

## Photothermal conversion property

Au@Ag$_2$Se-FA NPs were dispersed in DI water with the dispersion concentration ranging from 0 to 300 μg mL$^{-1}$. The dispersions (1 mL) at different concentrations were added in a quartz cuvette and irradiated by an optical fiber-coupled 808 nm high power diode-laser. The temperature rise induced by the NIR irradiation was monitored by a thermo-couple microprobe (Q50.5 mm) submerged in the dispersions. The photothermal conversion efficiency ($\eta$) of the NPs was calculated by the following equation: $\eta = \frac{hS(T_{max} - T_{surr}) - Q_s}{I(1 - 10^{-A_\lambda})}$ (details in Supplementary Table 1). The hearting-cooling curves were collected by irradiating the Au@Ag$_2$Se-FA dispersion (100 μg mL$^{-1}$, 1 mL, 808 nm, 1.2 W cm$^{-2}$) until reaching a steady-state temperature, followed by laser off and cooling to RT.

## Biocompatibility

To evaluate the cytotoxicity of Au@Ag$_2$Se-FA, HUVEC, and 4T1 cells were first seeded on a 96-well plate (1.0 × 10$^5$ cells per well) overnight, and then incubated with the NPs dispersions at various concentrations (0−200 μg mL$^{-1}$) for 24/48 h, respectively. The cell viability was measured using CCK-8 assay, and the optical absorbance at 450 nm of each well was measured using a microplate absorbance reader (BIO-680, USA).

The morphology of red blood cells (RBCs) and *E. coli* cells were imaged by scanning electron microscopy (SEM, JME-7500F). To evaluate the blood compatibility of Au@Ag$_2$Se-FA, the NPs dispersions (0.9 mL) at various concentrations (20−1000 μg mL$^{-1}$) were added in a centrifuge tube (1.5 mL) followed by addition of diluted RBCs dispersion (1 mL RBCs diluted in 10 mL PBS, 0.1 mL) and incubated at 37 °C for 24 h. RBCs incubated with DI water and PBS buffer (pH 7.4) were used as positive and negative control, respectively. The samples were centrifuged at 15,000 × *g* for 10 min, and the absorbance of their supernatants at 577 nm was measured. The diluted RBCs dispersion was then applied for SEM imaging.

## Evaluation of in vitro therapeutic effect

4T1 cells (1.0 × 10$^5$ cells per well) were seeded in a 96-well plate overnight, then incubated with Au@Ag$_2$Se-FA NPs dispersion (100 μg mL$^{-1}$). Next, the cells were irradiated by 808 nm laser (0.8 W cm$^{-2}$, 10 min), and the cell viability was evaluated using CCK-8 assay. To explore the therapeutic effect of CDT and chemotherapy of PI-Au@Ag$_2$Se-FA, Au@Ag$_2$Se-FA dispersions (20, 100, 200, 500, and 1000 μg mL$^{-1}$ in Dulbecco's modified Eagle's medium) were pre-irradiated (808 nm, 0.8 W cm$^{-2}$, 10 min) and incubated with 4T1 cells for 24 h. The cell viability treated with PI-Au@Ag$_2$Se-FA was finally determined using CCK-8 assay and fluorescence microscope. The cell cycle of PI-Au@Ag$_2$Se-FA (100 μg mL$^{-1}$)-treated 4T1 cells was analyzed using flow cytometry.

## qRT-PCR analysis

THP-1 cells were seeded in a 6-well plate (1.0 × 10$^6$ cells per well) overnight, then Phorbol-12-myristate-13-acetate (PAM, 1 mg mL$^{-1}$, 1 μL) was added to promote the transformation of THP-1 cells to M0 macrophages. Next, M0 macrophages were treated with (i) Au@Ag$_2$Se-FA NPs dispersion (100 μg mL$^{-1}$) (G1), (ii) PI-Au@Ag$_2$Se-FA dispersion (100 μg mL$^{-1}$) (G2), (iii) 808 nm laser (0.8 W cm$^{-2}$, 10 min) (G3), and (iv) Au@Ag$_2$Se-FA NPs (100 μg mL$^{-1}$) plus 808 nm irradiation (0.8 W cm$^{-2}$, 10 min) (G4), respectively, for 24 h. The cells in all groups were then collected for RNA isolation, and compared with the cells treated with 1640 medium or NIR irradiation alone. Total RNA was isolated using TRIzol reagent (Invitrogen, Carlsbad, USA) according to the manufacturer's instructions. First-strand cDNA was synthesized using the All-in-One First-Strand cDNA Synthesis Super Mix (TransGen, Beijing, China) for qRT-PCR analysis. The qRT-PCR analysis was carried out according to the manufacturer's instructions and the Perfect Start™ Green qPCR Super Mix (TransGen, Beijing, China) reaction set, with DI water as a negative control. The relative expression levels of the marker genes of M1-like (*TNFα*) and M2-like (*Arg-1*, *Fizz-1*) were calculated using the 2$^{-\Delta\Delta Ct}$ method[48,49]. All data were normalized to the levels of the internal control gene *Actb*. Each sample was quantified based on five biological replicates. The primers (Sangon Biotech, Shanghai) for these genes were listed in Supplementary Table 2. The gene expression difference between the treated and control groups was determined using Student's *t*-test; a gene with a relative expression ratio above the threshold (>1.5 or <0.66-fold) and *P* value < 0.05 was regarded as differentially expressed.

## In vitro antibacteria

*E. coli* cells (1.0 × 10$^5$ CFU) were incubated with (i) ceftriaxone, (ii) lobaplatin, (iii) Au@Cu$_{2-x}$Se, (iv) PI-Au@Cu$_{2-x}$Se, (v) PI-Au@Ag$_2$Se-FA and (vi) Au@Ag$_2$Se-FA, respectively, for 24 h. In all cases, the concentration of the agents was 100 μg mL$^{-1}$. For NP + NIR, the cells incubated with Au@Ag$_2$Se-FA NPs or Au@Cu$_{2-x}$Se NPs (100 μg mL$^{-1}$) were irradiated for 10 min (808 nm, 0.8 W cm$^{-2}$). After irradiation, the cells were washed three times with PBS, spread in Luria-Bertani semi-solid medium, and cultured for 24 h. SEM was used to explore the varied cell morphology after the treatments.

## Establishment of mice tumor model

1.0 × 10$^6$ 4T1 cells (or RM-1 cells) (50 μL in PBS) were subcutaneously injected into the right back of BALB/c mice. The mice injected with 4T1 cells were four-week-old female; those with RM-1 cells were four-to-six-week-old male (Charles River, Beijing). To construct the dual-tumor model, 1.0 × 10$^6$ 4T1 cells were subcutaneously injected into the right back of each mouse to establish the primary tumor, and 5.0 × 10$^5$ 4T1 cells were subcutaneously injected into the left back of each mouse to obtain the distal tumor. When the tumors grew to ~8 mm in diameter, the tumor-bearing mice were randomly divided into different groups for various treatments.

## Flow cytometric analysis of TME

After the mice were sacrificed, the tumors were collected and digested in dissociation buffer [mixture of deoxyribonuclease (50 μg mL$^{-1}$), hyaluronidase (100 μg mL$^{-1}$), and collagenase IV (2 mg mL$^{-1}$)] at 37 °C for 1 h with gentle shaking. The cell suspension was passed through a 70 μm cell strainer and dispersed in PBS buffer to form single-cell suspension. For T lymphocytes detection, the single-cell suspension was stained with antibodies against CD3, CD4 and CD8 to mark the helper (CD3$^+$ and CD4$^+$) and cytotoxic (CD3$^+$ and CD8$^+$) T cells. The tumor-cell suspension was stained with antibodies against CD11b, F4/80, CD86, and CD206 to analyze M1-like (CD11$^+$ and CD86$^+$) and M2-like (F4/80$^+$ and CD206$^+$) macrophages. The mice's blood and spleen were also collected for lymphocyte extraction using lymphocyte isolation kit (Solarbio) according to the manufacturer's manual. NK cells (CD3$^-$,

CD49b⁺), γδ T cells (CD3⁺, TCRβ⁻), and memory T cells (CD8⁺, CD44$^{high}$, CD62L$^{low}$) were analyzed using flow cytometer. The anti-mouse of CD3-PE (clone: 17A2), CD4-APC-cy7 (clone: GK1.5), CD8a-FITC (clone: 53-6.7), CD11b-PE (clone: M1/70), F4/80-APC (clone: BM8), CD86-PE-Cy7 (clone: GL-1), CD206-FITC (clone: C068C2), CD49b-APC (clone: HMα2), TCRβ-FITC (clone: H57-597), CD44-PE (clone: IM7), and CD62L-APC-Cy7 (clone: MEL-14) were purchased from Biolegend.

## Cytokine profile analysis

The cell supernatants, tumor tissues, and serum were collected after the treatments. The levels of IL-17, IL-10, TGF-β, PD-1, HSP-70, HMGB1, TNF-α, IL-12, IL-6, IL-1β, and IFN-γ were determined by enzyme-linked immunosorbent assay (ELISA). All ELISA kits were purchased from ZCIBIO Technology Co., Ltd.

## Evaluation of in vivo therapeutic effect

To evaluate the synergistic therapeutic effect of Au@Ag₂Se-FA on TRIM infection (or normal) tumor in vivo, after constructing 4T1 tumor model in the BALB/c mice, the tumor-bearing mice (tumor diameter of ~8 mm) were randomly divided into seven groups ($n = 5$ mice per group), where four of them (T2, T4, T6, T7) were *i.t.* injected with *E. coli* ($1 × 10^5$ *E. coli*, 50 μL in PBS). Three days later, T1 and T2 were *i.v.* injected with PBS (100 μL), T3–6 groups were *i.v.* injected with Au@Ag₂Se-FA NPs (5 mg mL⁻¹, 100 μL in PBS), and T7 was *i.t.* injected with Au@Cu₂₋ₓSe NPs (3 mg mL⁻¹, 50 μL in PBS). T5, T6, and T7 were irradiated at 12 h after *i.v.* injection (808 nm, 0.36 W cm⁻², 10 min). Upon irradiation, the body/tumor temperature was monitored using an IR thermal imager (Ti25, Fluke, USA). After the treatments, the weight and behavior of the mice and the volume of the tumors were recorded. The major organs were collected and stained using hematoxylin and eosin then visualized using an inverted fluorescence microscope (IX71, Olympus, Japan). After the observation at Day 16, five mice from each group were kept for survival analysis, and another nine mice from each group (T5 and T6) were re-challenged with 4T1 tumor cells ($5.0 × 10^5$) by subcutaneous administration (R-T5, R-T6). The survival analysis was repeated three times using three separate mice groups ($n = 5$ mice per group). Except those for survival analysis, the rest mice were sacrificed after all the experiments, following the Ethics Committee for Animal Experimentation guidelines. To investigate the influences of TRIM on breast tumors and immune regulation in TME, the tumor-bearing mice (tumor diameter of ~8 mm) were randomly divided into two groups. Mice in the EC group were *i.t.* injected with *E. coli* ($1 × 10^5$, 50 μL in PBS). The length and width of the mice's tumor were measured every two days after injection. Four mice from each group were randomly chosen and sacrificed at each time point (i.e., 0, 0.5, 3, 7, and 15 days after injection). The tumors were collected and three in each group were weighted. Under a sterile tactical environment, each tumor was cut into three portions (~100 mm³ each) for IHC, cytokine profile, and bacteria burden analysis. The tumor slices were deparaffinized, rehydrated, and then incubated with primary anti-CD3 (Ablonal, A1753, dilution 1:100), ant1-CD86 (Ablonal, A2352, dilution 1:100), anti-ARG-1 (Ablonal, A4923, dilution 1:100), and secondary antibodies (HRP Goat anti-rabbit IgG, Ablonal, AS014, dilution 1:100), followed by staining with DAB and hematoxylin. Fluorescent microscopy (Leica) was used to visualize the slides, and the data were analyzed using Image J.

## Bacteria burden in vivo

The bacteria burden of tumor was analyzed as follows: each tumor (0.1 g) was first mashed into tissue homogenate with sterile PBS (5 mL); diluted tissue homogenate (100 μL) was then uniformly covered on an agar plate and incubated at 37 °C for 24 h.

## Excretion study

Balb/c mice were *i.v.* injected with the Au@Ag₂Se-FA dispersion (50 mg mL⁻¹ in PBS, $n = 3$ mice per group). Four hours before each time point, the mice were housed in the cleansed metabolism cares to collect their urine. The collected urine was digested with aqua regia for ICP-MS analysis.

## Statistics and reproducibility

All experiments were conducted at least twice independently, yielding similar results. Replicates were reproducible. Biologically independent samples/animals per group and the experimental data were analyzed by one-way ANOVA with Tukey's post hoc test or two-tailed Student's *t*-test. $P < 0.05$ was considered to be statistically significant. Blinding was used in all biological experiments and no data were excluded from the analysis.

## Reporting summary

Further information on research design is available in the Nature Portfolio Reporting Summary linked to this article.

## Data availability

The authors declare that all data supporting the findings of this study are available within the paper [and its Supplementary Information]. Source data are provided with this paper.

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

## Acknowledgements

This work was financially supported by the National Natural Science Foundation of China (51772066, Y. S.; 52073074, Y. S.; 22272041, M. Y.), and State Key Laboratory of Urban Water Resource and Environment, Harbin Institute of Technology (2021TS08).

## Author contributions

Y.L.W., M.Y., and Y.S. designed the project; Y.L.W. and Y.Q.H. synthesized and characterized the samples; Y.L.W., C.H.Y., T.C.B., C.G.Z., and Z.T.W. carried out the in vitro/vivo experiments; Y.L.W., Y.S., M.Y., C.Y.C. and F.B. interpreted the results; Y.L.W., M.Y., Y.H. and Y.S. drafted/revised the manuscript; M.Y., Y.S and F.B. advised the project.

## Competing interests

The authors declare no competing interests.
