## [Peer Review File · Nature Communications]

REVIEWER COMMENTS

Reviewer #1 (Remarks to the Author):

Utilizing nanotechnology to manipulate the tumor associated microbiome is emerging as innovative strategies for cancer therapy. Recently, a large number of studies with ingenious designs have demonstrated the benefits of microbial regulation, including complete and targeted clearance, in tumor treatment, especially immunotherapy (Nat. Biomed. Eng., 2021, 5, 1377-1388; Nat. Biomed. Eng., 2022, 6, 32-43; Nat. Biomed. Eng., 2019, 3, 717–728; Adv. Mater., 2018, 30, 1805007). In this manuscript, the authors prepared folic acid modified Ag₂Se shell coated Au nanoparticles (Au@Ag₂Se) for multimodal antitumor therapy including PTT, CDT, chemotherapy and sterilization. Au@Ag₂Se showed strong in vitro sterilization on E. coli and an impressive relapse-free survival of > 500 days for 4T1 model treatment once combined with irradiation.

However, targeted regulation of tumor-resident intracellular microbiota (TRIM) in combination with other therapies is not as unique as the author claims, especially the synergistic immunotherapy based on Ag ion microbiota regulation has already been studied (Nat. Biomed. Eng., 2022, 6, 32-43). Despite the impressive in vivo results, I think the current manuscript can't reach the threshold of Nature communications.

Comments and suggestions for this manuscript.

1. The author claimed in the title, abstract, and introduction that Au@Ag₂Se can target TRIM, but there was a lack of detailed description on how to achieve targeted regulation and sufficient evidence to prove that.
2. The statement of "However, the impact of sterilization on tumor inhibition remains as an uncharted terrain" was not accurate, as it has been proved in several studies that thorough removal of intestinal microbiota can effectively delay the progress of tumors.
3. The authors should cite relevant literatures to confirm the effect of TRIM on cytokines secretion as they described in the section IV of the introduction.
4. CD86 is mainly expressed in immune cells including macrophages and dendritic cells. Only using CD86 could not accurately evaluate the expression levels of M1 macrophages in Supplementary Fig. 3.
5. It was very confused why the authors evaluated the release of HMGB1 in a mixed cells system of 4T1 and BMDMs after receiving different treatments, please supplement reasonable explanation. In addition, I wonder what kind of cells released HMGB1.
6. Whether different treatments are toxic to THP-1 cells?
7. Please provide explanations why these treatments directly functioned on THP-1 cells could affect the differentiation of THP-1 cells, and please provide results of changes in corresponding cellular pathway proteins.

8. The author claimed that Au@Ag₂Se treatment could restore anti-tumor immune surveillance, thus, the changes of the corresponding immune cells especially NK cells, $\gamma\delta$ T cells, and memory T cells after the treatment should be evaluated, as the cytokines level restored the baseline level in 7 days.
9. The residual amounts of E. coil in tumor tissues after different treatments should be evaluated.
10. The biodistribution of Au@Ag₂Se should be evaluated.
11. Intestinal microbiota has been proved to show a significant impact on the treatment of many solid tumors, including hepatic cancer and breast cancer. Thus, I suppose intravenously injected Au@Ag₂Se might impact gut microbiota which is also an issue that the author should pay attention to.
12. The gating strategies of the flow cytometry analysis should be provided.

Reviewer #2 (Remarks to the Author):

We appreciate the authors investigating this important topic and bringing forth the significance of an, often neglected, phenomenon which is the role of the tumor associate microbiota on both the progression and potential therapy of cancer. The authors emphasize that little to no work has been explored in combination sterilizing drugs/therapies with immunotherapies or chemotherapies for cancer treatment. Little is known about the potential of this combinatorial therapy for cancer and thus the thrust of the work investigates the issue using silver/selenide particles activated by near-infrared light.

Much has been published and understood in terms of the combination of hypothermia-inducing drugs or mechanisms and chemotherapy/immunotherapy. Hyperthermia indeed augments most cancer therapies and the authors note this point as well. Thus, it is not clear to this reviewer how the investigators deconvolute the role of sterilization (i.e. killing of bacterial pathogens) from hyperthermia in all the results showing enhanced responses with the nanoparticles. Hyperthermia is a byproduct of NIR activation and indeed may augment the sterilizing effect as well. In the absence of clear data delineating the role of hyperthermia versus sterilization (which can be induced by other means without hyperthermia), it is not clear if the conclusions regarding the sterilization effect are fully supported by the data. It certainly is helpful to have hyperthermia in the tumor microenvironment together with other therapies but this has been exhaustively in many different ways in prior publications with different cancer models.

Minor comment: Demonstrating the in vivo effects in at least two different cancer models will further bolster the generality of the therapeutic approach. In that event and in both cancer models, it needs to be understood from the data that the synergistic drug/immunotherapy and sterilizing effect is not partly or entirely due to additional mechanisms at work such as hyperthermia in the tumor microenvironment.

Along the line of deconvoluting the particle/NIR effects, it would be important to include cytokine data from healthy animals that have undergone particle/NIR therapy. This is to ensure that the composite particle is not upregulating or downregulating key immune signals. Pro and anti-inflammatory cytokine analysis from draining lymph nodes would suffice for this purpose.

Reviewer #3 (Remarks to the Author):

The work by Wang et al. addressed novel and important research questions on the role of tumor-resident intracellular microbiota (TRIM) in immunosuppressive tumor microenvironment. Further, they have reported a promising single nanotherapeutic formulation, which exhibits photothermal, chemodynamic, chemotherapeutic, and bacteria-killing properties simultaneously upon NIR. The impact and interest of research question deems appropriate for Nature Communications. However, the contribution of TRIM and sterilization in the tumor appears small, and more data are needed to strengthen the story. For example, bacteria infection in mice (without tumors) is needed to tease out anti-bacteria immunity and anti-cancer immunity. Standard antibiotic (ceftriazone) appears to have minimal effect in controlling tumors (e.g., in NS2 group). Further, the difference in efficacy of sterilizing NP vs. non-sterilizing NP also appears to be small (e.g., T6 vs. T7). Longer-term survival profile of T6 vs T7 will be needed to provide more convincing evidence. Other following issues on premise, study design, data reporting, and statistics also need to be addressed before the reviewer could recommend the article for publication.

1. Issues on the premise regarding TRIM in cancer and the proposed nanotherapeutic.

a) Intratumoral injection of E.coli could trigger a lot of host's immune response against bacteria itself (especially innate immunity). Some of the observed effects in cytokine levels could be bacteria-related and not tumor-related. Current data do not provide insight on this yet. Data on mouse injected with bacteria alone is needed.

b) Injecting bacteria to tumor could promote ulceration, and tumor volume alone is not a good measure of tumor burden. Additional analysis of tumor burden is needed to substantiate this.

c) Justification of 500,000 E. coli cells used in the tumor model is needed. How representative is this to real patient scenario?

d) The authors assessed impact of TRIM on tumor's immune milieu by IHC analysis of CD3, CD86, and ARG. The authors claimed CD86 to represent M1 macrophages and ARG to represent M2 macrophages. This is not correct because CD86 is also expressed in DC, monocytes, and T cells. ARG (presumable ARG-1 – please clarify) is also expressed in other myeloid cells (e.g., MDSC). The flow analysis that the author later did is more appropriate. However, that presented dataset is not complete (only T2, T6 and T7 data are shown in Fig. S25). The authors should present all flow data for T1-T7. Flow cytometry of T1 vs. T2 will be more convincing in showing the role of TRIM in tumor's immune milieu.

e) Metastasis aspect of the disease was not discussed or studied. NIR is limited to superficial solid tumors. It is important to understand if this treatment could sufficiently curb metastasis diseases or this technology will only benefit localized disease. For example, the triggered immunity by NPs+NIR could potentially affect distal or metastatic tumors (which will not be irradiated). Additional study showing abscopal effect of NPs+NIR will substantially enhance the impact of the work.

f) Does low efficacy of ceftriazone alone (NS2) in Fig.6 and Fig. S27 suggest small contribution of sterilization to tumor therapy because the author mentioned ceftriazone as being a powerful antibiotic in clinic already? Lobaplatin (used in NS3 and NS4 group) has cancer-killing effect (shown in Fig. S26), so it does not show sterilization effect alone. Please address/discuss this finding more in Discussion section.

g) The impact on T cells was discussed and reported. However, it is unclear if the generated T cells is tumor-specific or bacteria-specific. Evidence of tumor-specific immune response could be provided by either tetramer staining of splenocytes, or an ex vivo stimulation assay of splenocytes with tumor lysate or relevant tumor antigens, or an in vivo tumor rechallenge study (where tumors are implanted to the cured mice months after treatment).

h) What happens to the released Au, Ag, and Se in the body? Short-term study reveals no effects in livers and kidneys. However, long-term effect or deposition of these elements to organs is not described. It is unclear how many % of what's injected can be recovered in urine cumulatively (e.g., XX% of administered Au, Ag, and Se is recovered in urine by YY days). Please perform mass balance analysis (with data from Fig. S22) to address this question.

i) Treg, which is part of CD3+CD4+, is not measured or addressed. CD8/Treg is another important indicator beyond CD8 T cells alone.

2. Issues on material characterization and justification of folic acid

j) The authors did a great job on establishing 'identity' of the synthesized material. However, composition analysis is not adequate yet. For example, how much of folic acid got loaded on the nanoparticle and how did the authors measure it? Also, in the method section, there appears to be no purification after folic acid loading. Please clarify and elaborate. DLS data (hydrodynamic size) of NP is currently presented in the supplementary data. However, the size of NP in solution and biological matrix (e.g., serum) is important for nanomedicine. The authors should thus at least mention Z-average and PDI in the main paper.

k) Please confirm that ALL studies in the paper that use NPs (e.g., characterization, stability, Ag+ release) employed the version with folic acid. If no, please provide distinguishing terms between NP with vs. without folic acid. If yes, please include one sentence confirming this in the paper (e.g., "Au@Ag2Se (which includes folic acid) is used in all studies").

l) Has the importance and necessity of folic acid been experimentally shown and proven for this specific nanoformulation yet (e.g., data of NP with vs. without folic acid)? Please discuss and include relevant data.

3. Issues on study designs (missing controls), data presentation, and data interpretation

- m) Immune response in vitro needs a PI-Au@Ag₂Se control because that would rule out hyperthermia effect in the observed phenotype. The justification/relevance of M0 macrophage model is not described. THP-1 cell was not mentioned outside method section.
- n) Fig. S12 shows a much more pronounced effect of PI-Au@Ag₂Se over Au@Ag₂Se+NIR. Please discuss. Fig. S16 doesn't have NPs+NIR, which is a key group.
- o) Fig. S21b should be of the same format as Fig. S21a to maintain coherence. In particular, x-axes should be biomarkers consistently (e.g., AST, ALP, ALT, CREA, UA, BUN).
- p) Fig. S14 [100 ug/ml Au@Ag₂Se] shows less kill than the same data in Fig. 4B. Please explain why? Clearer and more detailed method section throughout the paper (see section 4) may clarify.
- q) Efficacy of T6 vs T7 is small, but appears significant. Long-term survival of T7 (in the same manner as Fig. 5C) should be provided to show clear benefits over T6.
- r) 'mRNA level' should be spelled out, instead of 'gene expression level'.
- s) Explanation of Fig. S25 in text is not complete yet. The authors discussed T cell populations for Au@Ag₂Se and then jumped to discuss macrophage populations for Au@Cu₂-xSe₂. Please revise. In another sentence on NS1's cytokine measurement, Fig. 5d seems incorrectly introduced. Please fix or clarify its inclusion as needed.
- t) Please consider having a separate session on non-sterilizing nanoparticle, such that 'Discussion' section does not contain a lot of newly introduced data.

4. Issues on incomplete method descriptions

- u) It is unclear how the authors monitor bacteria burden in the tumor. Please explain more. The reviewer couldn't find such info in method or supplementary section yet.
- v) Please provide information on all antibody clones, conjugated dyes, vendors, dilution factor, used in all assays.
- w) Method section does not describe sample preparation and staining protocols for IHC (unlike flow cytometry). Further, it is unclear how many images and their field of view (e.g., how many mm³) were being analyzed for IHC per group. Some representative images in the supplementary info do not agree with certain timepoints in Fig. 2.
- x) Justification of co-culturing 4T1 and BMDM to monitor HMGB release is not described. Will 4T1 still produce HMGB in the absence of BMDM?
- y) Certain other elements are missing in the Method section. Please provide more info throughout such that the work can be reproduced in the future.
 - For example, sample preparation method for SEM imaging is not provided. It is unclear what diluted RBCs dispersion means (and how it's prepared).

- Cell number for 4T1 in in vitro therapeutic effect is missing.
- Mouse model and treatment deserves its own section. Currently, they are mentioned across two sections in the method. The readers need to combine info across two sections to understand the experiment. Timeline for all in vivo studies need clarification (e.g., how many days after implantation did the treatment start? mouse age? mouse vendor?).
- For in vivo efficacy, was the endpoint set at 16 days post-treatment? were separate studies with another set of mice conducted for survival analysis? Or were they conducted at the same time and a subset of mice were sac'ed at 16 days?

5. Issues on rigor and Statistics

z) Statistic analysis is not performed for tumor growth curve and survival. These need different analyses beyond one-way ANOVA.

Reviewer: #1

Comments:

Utilizing nanotechnology to manipulate the tumor associated microbiome is emerging as innovative strategies for cancer therapy. Recently, a large number of studies with ingenious designs have demonstrated the benefits of microbial regulation, including complete and targeted clearance, in tumor treatment, especially immunotherapy (Nat. Biomed. Eng., 2021, 5, 1377-1388; Nat. Biomed. Eng., 2022, 6, 32-43; Nat. Biomed. Eng., 2019, 3, 717–728; Adv. Mater., 2018, 30, 1805007). In this manuscript, the authors prepared folic acid modified Ag₂Se shell coated Au nanoparticles (Au@Ag₂Se) for multimodal antitumor therapy including PTT, CDT, chemotherapy and sterilization. Au@Ag₂Se showed strong in vitro sterilization on E. coil and an impressive relapse-free survival of > 500 days for 4T1 model treatment once combined with irradiation. However, targeted regulation of tumor-resident intracellular microbiota (TRIM) in combination with other therapies is not as unique as the author claims, especially the synergistic immunotherapy based on Ag ion microbiota regulation has already been studied (Nat. Biomed. Eng., 2022, 6, 32-43). Despite the impressive in vivo results, I think the current manuscript can't reach the threshold of Nature communications.

Answer: We are grateful to the reviewer for taking the time to evaluate our manuscript, raising constructive comments, and emphasizing the importance of our topic.

To address this reviewer's concern, we have identified the following major discrepancies between [Nat. Biomed. Eng., 6, 32-43 (2022), R1] mentioned by the reviewer and our work:

- *Different strategies for tumor inhibition*

R1: A triple combination of drugs was used, including i) hydrogel-embedded silver nanoparticles with ii) exogenous competing bacteria (*P. anaerobius*) and iii) an immune checkpoint inhibitor (anti-PD-1).

This work: We used only a single Au@Ag₂Se agent, with no immune-stimulating drugs (e.g., immune modulators, immune checkpoint blockades, or antigens) or any antitumor probiotics.

- *The opposite role of bacteria in immune regulation*

R1: Neither immunosuppression nor tumor overgrowth induced by tumor-resident intracellular microbiota (TRIM) was revealed. The effects of TRIM on tumors were unclear. The exogenous bacteria were used to promote tumor inhibition and immune responses. The introduction of exogenous bacteria rather than sterilization contributed primarily to the antitumor efficacy.

This work: We demonstrated TRIM-induced immunosuppression and tumor overgrowth in both breast and prostate carcinoma. We revealed the crucial role of sterilization in immune regulation and tumor inhibition.

- *Effect of Ag-based agents on immune regulation*

R1: Ag gel alone had insignificant effects on immune regulation, indicating that the sterilization induced by the given Ag agent cannot provide a sufficient immunological effect. For the combination of the Ag gel with the exogenous bacteria and immune checkpoint inhibitor, it is difficult to separate the role of sterilization

from the complicated influence delivered by the latter two.

This work: Significant tumor inhibition and immunological activation were accomplished by utilizing a single Au@Ag₂Se. By comparing Au@Ag₂Se with Au@Cu_{2-x}Se, we deconvoluted the role of sterilization and validated its importance in antitumor therapeutic efficacy and immune responses.

- *Different efficacy for tumor inhibition*

R1: When mice were treated with the Ag gel alone, they began to die within 20 days post-treatment, and >80% died within 40 days post-treatment; when mice were treated with the combination of Ag gel, *P. anaerobius* and anti-PD-1, mice barely survived 70 days post-treatment.

This work: A relapse-free survival of >700 days is achieved by our single agent.

- *Immunological memory and systemic immune responses*

R1: It showed no results in terms of immunological memory or systemic immune responses.

This work: We demonstrated the presence of long-lasting immunological memory and systemic immune responses by secondary inoculation and distal tumor studies.

- *Cancer types and meanings*

R1: A single oral squamous cell carcinoma.

This work: Both breast and prostate carcinoma, representing more general cancer types and hardly treated by injection with exogenous bacteria.

1. The author claimed in the title, abstract, and introduction that Au@Ag₂Se can target TRIM, but there was a lack of detailed description on how to achieve targeted regulation and sufficient evidence to prove that.

Answer: We appreciate the reviewer raising this point. In the present system, we achieved tumor-resident intracellular microbiota (TRIM) targeting by the tumor-targeting of Au@Ag₂Se-FA nanoagent. To enable tumor targeting, folic acid (FA) was equipped on the NPs' surface. As a result, a pronounced tumor targeting effect was demonstrated: measured by inductively coupled plasma mass spectroscopy, a high Au ion level of ~17.8% ID·g⁻¹ was detected from the tumor area at 12 h post-injection; the excellent photothermal and antibacterial properties accomplished by intravascular injection of the nanoagent also suggested efficient accumulation of the nanoagent at the tumor area. Furthermore, the bacterial content in the tumor after the various treatments verified the successful TRIM-targeted sterilization by Au@Ag₂Se-FA upon near-infrared irradiation. Following this reviewer's suggestions, we conducted additional experiments and included the results and associated descriptions in the revised manuscript (Pages 5, 7, 8, and 10) and Supplementary Information (SI) as follows:

"To enable tumor targeting, FA was equipped on the NPs' surface (Fig. 1 and Supplementary Fig. 5), and the resultant Au@Ag₂Se-FA NPs were used in all studies."

"The biodistribution of Au@Ag₂Se-FA was assessed by measuring the content of Au ions in the major organs of the tumor-bearing BALB/c mice at 12 h after i.v. injection. A high level of ~17.8% ID·g⁻¹ was detected from the tumor area, indicating efficient uptake of Au@Ag₂Se-FA by the tumor (Supplementary Fig. 20). The agent's

tumor-targeting accumulation was also verified by the substantial tumor-selective photothermal effect after i.v. injection of Au@Ag₂Se-FA (Supplementary Fig. 21), where the surface modification of FA promoted the tumor-targeting effect.”

Supplementary Fig. 20 | Biodistribution of Au@Ag₂Se-FA in the major organs 12 h after i.v. injection, determined by the concentration of Au ions in the tissue lysates using ICP-MS (n = 3).

Supplementary Fig. 21 | In vivo IRT images of 4T1 tumor-bearing mice. Whilst the tumor temperature of mice treated by i.v. injection of Au@Ag₂Se is < 46 °C upon 10 min NIR irradiation (808 nm, 0.36 W·cm⁻²), the tumor temperature of mice treated by i.v. injection of Au@Ag₂Se-FA quickly increases to ~55 °C upon the same irradiation, suggesting the enhanced tumor-targeting effect by FA modification.

“The residual *E. coli* level in T6 tumor tissues was examined (Supplementary Fig. 25), confirming the potent sterilization of NPs+NIR.”

Supplementary Fig. 25 | **a**, Representative photos for bacteria culture analysis and **b**, bacteria density measured from the breast tumors of T2, T6, T7 after various treatments (n = 3).

“The tumor-bearing mice raised in the non-sterile environment were divided into five groups (n = 5), including i) the

untreated group (NS1), ii) the mice treated by ceftriaxone alone (NS2), iii) by lobaplatin alone (NS3), iv) by the combination of ceftriaxone and lobaplatin (NS4), and v) by Au@Ag₂Se-FA plus irradiation (NS5). ...The NS2, NS4, and NS5 groups showed strong sterilization effect (Supplementary Fig. 38).”

Supplementary Fig. 38 | a, Representative photos for bacteria culture analysis, and **b**, bacteria density measured from the breast tumors of the NS1, NS2, NS3, NS4, and NS5 groups after the different treatments (n = 3).

2. The statement of “However, the impact of sterilization on tumor inhibition remains as an uncharted terrain” was not accurate, as it has been proved in several studies that thorough removal of intestinal microbiota can effectively delay the progress of tumors.

Answer: We thank the reviewer for raising this point. Following this reviewer’s suggestion, we have made the following revisions in the revised manuscript (Page 3):

“Significant progress has been made in enhancing anticancer efficacy and immunity responses by regulating intestinal microbiota [10,11]. Still, the impact of sterilization in TME on tumor inhibition remains unclear.”

[10] Simpson, R.C. et al. Towards modulating the gut microbiota to enhance the efficacy of immune-checkpoint inhibitors. *Nat. Rev. Clin. Oncol.* **20**, 697–715 (2023).

[11] Lang, T. et al. Combining gut microbiota modulation and chemotherapy by capecitabine-loaded prebiotic nanoparticle improves colorectal cancer therapy. *Nat. Commun.* **14**, 4746 (2023).

3. The authors should cite relevant literatures to confirm the effect of TRIM on cytokines secretion as they described in the section IV of the introduction.

Answer: We thank the reviewer for raising this comment. Following the reviewer’s suggestion, we have added the associated references in Section IV of Introduction in the revised manuscript (Page 4) as follows:

“Escherichia coli (E. coli), a Proteobacteria strain abundant in both malignant and normal breast cells [4,5], is employed to construct the TRIM model. The TRIM has been found to up-regulate the expression of immunosuppressive cytokines [interleukin-10 (IL-10), transforming growth factor-β (TGF-β)] and pro-inflammatory cytokines IL-17, while down-regulating the expression of pro-inflammatory cytokines [including IL-12, tumor necrosis factor-α (TNF-α), and interferon-γ (IFN-γ)] and programmed cell death-1 (PD-1) cytokines [12,28].”

[12] Pushalkar, S. et al. The pancreatic cancer microbiome promotes oncogenesis by induction of innate and adaptive immune suppression. *Cancer Discov.* **8**, 403–416 (2018).

[28] Lacet, R. T., Bleich, R. M. & Arthur, J. C. Microbia effects on carcinogenesis: Initiation, promotion, and progress. *Annu. Rev. Med.* **72**, 243–261 (2021).

4. *CD86 is mainly expressed in immune cells including macrophages and dendritic cells. Only using CD86 could not accurately evaluate the expression levels of M1 macrophages in Supplementary Fig.*

Answer: We are grateful to the reviewer for raising this constructive comment. Following this reviewer's suggestion, we performed additional experiments by flow cytometric analysis to evaluate the M1 macrophages. We have added the results and associated descriptions in the revised manuscript (Page 5) and SI as follows:

“Furthermore, as revealed by the immunohistochemistry (IHC) analysis (Fig. 2f and Supplementary Fig. 3), the numbers of T cells (CD3⁺) and M2 TAMs (ARG-1⁺) evidently varied after injection: at Day 15, the number of T cells reduced to 27%, while M2 TAMs increased to 286%. In contrast, these cell numbers of the control group were barely varied. Consistently, flow cytometric analysis (Supplementary Fig. 4) demonstrated that E. coli injection reduced the number of M1 TAMs (CD11b⁺, CD86⁺) to 65%.”

Supplementary Fig. 4 | a, Flow cytometric gating strategy for M1 TAMs (CD11b⁺, CD86⁺) panel of Control and EC. **b**, The number of CD11b⁺ and CD86⁺ M1 TAMs per 50,000 cells of tumors in Control and EC (n = 4). Data are presented as mean ± standard deviation. One-way ANOVA with Tukey's post hoc test. ** P < 0.01.

5. *It was very confused why the authors evaluated the release of HMGB1 in a mixed cells system of 4T1 and BMDMs after receiving different treatments, please supplement reasonable explanation. In addition, I wonder what kind of cells released HMGB1.*

Answer: We thank the reviewer for raising this comment. In fact, co-culturing the intended cancer cells with BMDM is a commonly-used approach to monitor HMGB1 release upon the phagocytic clearance of macrophages [*Nat. Biomed. Eng.* **4**, 1102–1116 (2020)]. Similar to the earlier results [*Nat. Med.* **13**, 1050–1059 (2007)], HMGB1 was released from the dying 4T1 cells in the present case. Following this reviewer's suggestion, we have the following sentences in the revised manuscript (Page 7):

“Next, we cultured 4T1 cells with bone marrow-derived macrophages (BMDMs) (Fig. 4d) to evaluate the release of high mobility group box1 (HMGB1) upon macrophage phagocytic clearance [22]. Au@Ag₂Se-FA alone could not regulate HMGB1. The high HMGB1 level of NPs+NIR (464% as that of Control) indicated secondary necrosis and enhanced immune responses [37,38] even upon macrophage clearance.”

Fig. 4 | Therapeutic efficacy and immune response induced by Au@Ag₂Se in vitro. **d**, MGB1 level in the supernatants of 4T1 cells after various treatments, including i) 4T1 cells co-cultured with BMDMs for 12 h (Control), ii) 4T1 cells culture with Au@Ag₂Se-FA for 12 h then co-cultured with BMDMs for another 12 h (Au@Ag₂Se-FA), iii) 4T1 cells treated by NPs+NIR and co-cultured with BMDMs for 12 h (NPs+NIR).

6. Whether different treatments are toxic to THP-1 cells?

Answer: We appreciate the reviewer raising this important question. Following this reviewer’s suggestion, we have evaluated the toxicity of the different treatments to THP-1 cells and added the results and associated descriptions in the revised manuscript (Page 7) and SI as follows:

“Human leukemia monocyte THP-1 cells were treated with 12-O-tetradecanoylphorbol-13-acetate to get M0 macrophages [39]. After being treated with i) Au@Ag₂Se-FA alone (G1), ii) PI-Au@Ag₂Se-FA alone (G2), iii) NIR irradiation only (G3), and iv) Au@Ag₂Se-FA plus NIR irradiation (G4), all groups showed high cell viability (> 93.7%), suggesting rather low/no cytotoxicity to M0 macrophages (Supplementary Fig. 17).”

Supplementary Fig. 17 | Cell viability of M0 macrophages after treated with i) pristine Au@Ag₂Se-FA (100 μg·mL⁻¹) (G1), ii) PI-Au@Ag₂Se-FA (100 μg·mL⁻¹) (G2), iii) NIR irradiation only (0.8 W·cm⁻², 10 min) (G3), and iv) Au@Ag₂Se-FA plus NIR irradiation (G4), with the untreated M0 macrophage cells as Control.

7. Please provide explanations why these treatments directly functioned on THP-1 cells could affect the differentiation of THP-1 cells, and please provide results of changes in corresponding cellular pathway proteins.

Answer: We thank the reviewer for raising this comment. As previously demonstrated [Research 2022, 9854904 (2022)], M0 macrophages can be obtained by treating human leukemia monocyte THP-1 cells with 12-O-tetradecanoylphorbol-13-acetate for 24 h. The variation of M0 macrophage cells after the treatments is closely associated with the level of reactive nitrogen and oxygen species (RNS and ROS) in M0 macrophages. Following this reviewer’s suggestion, we have carried out additional experiments and included the following results and associated descriptions in the revised manuscript (Page 7) and SI:

“Reactive nitrogen species (RNS) and ROS were largely promoted in M0 macrophages of G2 and G4, with RNS being 287% and 284% and ROS being 202% and 195% as that of Control, respectively (Supplementary Fig. 18). High ROS and RNS levels can act as second messengers in macrophages, stimulating pro-inflammatory signaling cascades and promoting M1 transformation [40]. This was supported by the up-regulated expression of IL-12 in G2 and G4 (Supplementary Fig. 19).”

Supplementary Fig. 18 | a, Nitric oxide (NO) and **b**, ROS release from M0 macrophages by i) Au@Ag₂Se-FA (100 µg·mL⁻¹) without irradiation (G1), ii) PI-Au@Ag₂Se-FA (100 µg·mL⁻¹) (G2), iii) NIR irradiation only (0.8 W·cm⁻², 10 min) (G3), and iv) Au@Ag₂Se-FA plus NIR irradiation (G4) (n = 5). Data are presented as mean ± standard deviation. One-way ANOVA with Tukey’s post hoc test. **** P < 0.001.

Supplementary Fig. 19 | IL-12 level in the supernatants from M0 macrophages treated with i) Au@Ag₂Se-FA (100 µg·mL⁻¹) (G1), ii) PI-Au@Ag₂Se-FA (100 µg·mL⁻¹) (G2), iii) NIR irradiation only (0.8 W·cm⁻², 10 min) (G3), and iv) Au@Ag₂Se-FA plus NIR irradiation (G4) (n = 5). Data are presented as mean ± standard deviation. One-way ANOVA with Tukey’s post hoc test. ** P < 0.001.**

8. The author claimed that Au@Ag₂Se treatment could restore anti-tumor immune surveillance, thus, the changes of the corresponding immune cells especially NK cells, γδ T cells, and memory T cells after the treatment should be evaluated, as the cytokines level restored the baseline level in 7 days.

Answer: We highly appreciate the reviewer’s constructive comment. Following this reviewer’s suggestions, we have carried out additional experiments to evaluate the NK cells, γδ T cells, and memory T cells by flow cytometric analysis after the different treatments. We have included the results and associated descriptions in the revised manuscript (Page 9) and SI as follows:

“To demonstrate the long-term durability of the immune responses, 45 days after the treatment, mice of T5 and T6 (n = 9) that had survived from the first inoculation of 4T1 tumors were re-challenged via subcutaneous administration of 4T1 tumor cells (1.0 × 10⁶) at the previously tumor-bearing area without any additional treatments (R-T5 and R-T6 groups). 10 days after the re-challenge, the mice’s blood and spleens were evaluated by flow cytometric analysis (Fig. 5f, Supplementary Figs. 27–29). Excitingly, all these mice still exhibited significant immune responses. The number of NK cells (CD3⁻, CD49b⁺) [42] in R-T5 and R-T6 grew to 455% and 481%, the γδ T cells (CD3⁺, TCRβ⁻) [43] increased to 363% and 389%, and the memory T cells (CD8⁺, CD44^{high}, CD62L^{low}) [22] increased to 601% and 606%, as compared to the untreated tumor-bearing mice (Control). Both R-T5 and R-T6 showed no sign of tumors 21 days after the re-challenge (Supplementary Fig. 30). The results emphasized the long-lasting immunological memory against 4T1 malignancy by NPs+NIR treatment. Such memory may have played a crucial role in the complete cure.”

Fig. 5 | Therapeutic effect of Au@Ag₂Se-FA in vivo. f, Flow cytometric analysis for the number of NK cells (CD3⁻, CD49b⁺), γδ T cells (CD3⁺, TCRβ⁻), and memory T cells (CD8⁺, CD44^{high}, CD62L^{low}) per 100,000 cells in blood or spleen of the R-T5 and R-T6 groups, with the results from the untreated 4T1 tumor-bearing mice as Control (n = 5).

Supplementary Fig. 27 | Representative flow cytometric gating strategy for NK cells (CD3⁻, CD49b⁺) panel in blood of the R-T5, R-T6, and control groups.

Supplementary Fig. 28 | Representative flow cytometric gating strategy for $\gamma\delta$ T cells ($CD3^+$, $TCR\beta^-$) panel in blood of the R-T5, R-T6, and control groups.

Supplementary Fig. 29 | Representative flow cytometric gating strategy for memory T cells ($CD8^+$, $CD44^{high}$, $CD62L^{low}$) panel in spleens of the R-T5, R-T6, and control groups.

Supplementary Fig. 30 | Digital photo (tumors in Control at Day 21) and b, mean weight of the tumors collected from the mice 21 days after the various treatments ($n = 4$).

9. The residual amounts of *E. coli* in tumor tissues after different treatments should be evaluated.

Answer: We appreciate the reviewer raising this valuable comment. Following this reviewer's suggestion, we have performed additional experiments and included the results and associated descriptions in the revised manuscript (Pages 8 and 10) and SI as follows:

"The residual E. coli level in T6 tumor tissues was examined (Supplementary Fig. 25), confirming the potent sterilization of NPs+NIR."

Supplementary Fig. 25 | a, Representative photos for bacteria culture analysis and **b**, bacteria density measured from the breast tumors of T2, T6, T7 after various treatments ($n = 3$).

“The tumor-bearing mice raised in the non-sterile environment were divided into five groups ($n = 5$), including i) the untreated group (NS1), ii) the mice treated by ceftriaxone alone (NS2), iii) by lobaplatin alone (NS3), iv) by the combination of ceftriaxone and lobaplatin (NS4), and v) by Au@Ag₂Se-FA plus irradiation (NS5). ...The NS2, NS4, and NS5 groups showed strong sterilization effect (Supplementary Fig. 38).”

Supplementary Fig. 38 | a, Representative photos for bacteria culture analysis, and **b**, bacteria density measured from the breast tumors of the NS1, NS2, NS3, NS4, and NS5 groups after the different treatments ($n = 3$).

10. The biodistribution of Au@Ag₂Se should be evaluated.

Answer: We appreciate the reviewer’s constructive comments. Following the reviewer’s suggestion, we have evaluated the biodistribution of Au@Ag₂Se-FA and included the results and associated descriptions in the revised manuscript (Page 7) and SI as follows:

“The biodistribution of Au@Ag₂Se-FA was assessed by measuring the content of Au ions in the major organs of the tumor-bearing BALB/c mice at 12 h after i.v. injection. A high level of ~17.8% ID·g⁻¹ was detected from the tumor area, indicating efficient uptake of Au@Ag₂Se-FA by the tumor (Supplementary Fig. 20). The agent’s tumor-targeting accumulation was also verified by the substantial tumor-selective photothermal effect after i.v. injection of Au@Ag₂Se-FA (Supplementary Fig. 21), where the surface modification of FA promoted the tumor-targeting effect. The levels in mononuclear phagocyte system organs, i.e., liver, spleen, and kidney, were 31.6% ID·g⁻¹, 10.1% ID·g⁻¹, and 7.5% ID·g⁻¹, respectively.”

Supplementary Fig. 20 | Biodistribution of Au@Ag₂Se-FA in the major organs 12 h after i.v. injection, determined by the concentration of Au ions in the tissue lysates using ICP-MS (n = 3).

11. Intestinal microbiota has been proved to show a significant impact on the treatment of many solid tumors, including hepatic cancer and breast cancer. Thus, I suppose intravenously injected Au@Ag₂Se might impact gut microbiota which is also an issue that the author should pay attention to.

Answer: We appreciate the reviewer raising this interesting point. Following this reviewer’s suggestion, we have also measured the content of Au@Ag₂Se-FA in the gut at 12 h after i.v. injection and evaluated the impact of the agent on the intestinal probiotics (*Lactobacillus reuteri* and *Bifidobacterium longum*). We have included the results and associated descriptions in the revised manuscript (Page 9) and SI as follows:

“The agent content in the intestine was detected to be 6.7% ID·g⁻¹ in terms of Au ions, corresponding to 56 μg·mL⁻¹ of Au@Ag₂Se-FA. Even at a higher concentration (100 μg·mL⁻¹), Au@Ag₂Se-FA had no impact on the intestinal probiotics *Lactobacillus reuteri* and *Bifidobacterium longum* (Supplementary Fig. 32).”

Supplementary Fig. 32 | Absorbance of **a**, *Lactobacillus reuteri* and **b**, *Bifidobacterium longum* suspension at 600 nm after co-incubating the bacteria with Au@Ag₂Se-FA (100 μg·mL⁻¹) or PBS (Control).

12. The gating strategies of the flow cytometry analysis should be provided.

Answer: We appreciate the reviewer raising this important point. Following the reviewer’s suggestion, we have provided the gating strategies of the flow cytometric analysis in SI as follows:

Supplementary Fig. 27 | Representative flow cytometric gating strategy for NK cells ($CD3^{-}$, $CD49b^{+}$) panel in blood of the R-T5, R-T6, and control groups.

Supplementary Fig. 28 | Representative flow cytometric gating strategy for $\gamma\delta$ T cells ($CD3^{+}$, $TCR\beta^{-}$) panel in blood of the R-T5, R-T6, and control groups.

Supplementary Fig. 29 | Representative flow cytometric gating strategy for memory T cells ($CD8^{+}$, $CD44^{high}$, $CD62L^{low}$) panel in spleens of the R-T5, R-T6, and control groups.

Supplementary Fig. 42 | Representative flow cytometric gating strategy for help ($CD3^{+}$, $CD4^{+}$) and cytotoxic ($CD3^{+}$, $CD8^{+}$) T cells panel in the tumors of the RM-C, RM-T2, RM-T3, and RM-T4 groups three days after the various treatments. The RM-T2 and RM-T4 groups were treated by *i.t.* injection of *E. coli* three days before the treatments. RM-T2 had no subsequent treatment. RM-T3 and RM-T4 were treated by NIR irradiation 12 h after *i.v.* injection of Au@Ag₂Se-FA. RM-C was the control group without any treatment.

Supplementary Fig. 43 | Representative flow cytometric gating strategy for M1 (CD86⁺, CD11b⁺) and M2 (CD206⁺, F4/80⁺) TAMs panel in the tumors of the RM-C, RM-T2, RM-T3, and RM-T4 groups three days after the various treatments.

Reviewer: #2

Comments:

We appreciate the authors investigating this important topic and bringing forth the significance of an, often neglected, phenomenon which is the role of the tumor associate microbiota on both the progression and potential therapy of cancer. The authors emphasize that little to no work has been explored in combination sterilizing drugs/therapies with immunotherapies or chemotherapies for cancer treatment. Little is known about the potential of this combinatorial therapy for cancer and thus the thrust of the work investigates the issue using silver/selenide particles activated by near-infrared light.

Answer: We are grateful to the reviewer for taking the time to evaluate our manuscript and providing positive remarks and constructive comments. Following this reviewer’s suggestions, we have carried out additional experiments and included the new results in the revised manuscript and SI. Please refer to our point-by-point responses below.

---Much has been published and understood in terms of the combination of hypothermia-inducing drugs or mechanisms and chemotherapy/immunotherapy. Hyperthermia indeed augments most cancer therapies and the authors note this point as well. Thus, it is not clear to this reviewer how the investigators deconvolute the role of sterilization (i.e. killing of bacterial pathogens) from hyperthermia in all the results showing enhanced responses with the nanoparticles. Hyperthermia is a byproduct of NIR activation and indeed may augment the sterilizing effect as well. In the absence of clear data delineating the role of hyperthermia versus sterilization (which can be induced by other means without hyperthermia), it is not clear if the conclusions regarding the sterilization effect are fully supported by the data. It certainly is helpful to have hyperthermia in the tumor microenvironment

together with other therapies but this has been exhaustively in many different ways in prior publications with different cancer models.

Answer: We highly appreciate the reviewer raising this important point.

To deconvolute the role of Ag ions on bacterial pathogens from that of hyperthermia in sterilization, we used a comparable agent, Au@Cu_{2-x}Se (Au core covered by Cu_{2-x}Se shell). Au@Cu_{2-x}Se had a similar morphology as that of Au@Ag₂Se; it can serve photothermal conversion and chemical dynamic therapy (PTT and CDT) like Au@Ag₂Se but without inherent antibiotic function. As seen in Supplementary Fig. 16, Au@Cu_{2-x}Se showed no sterilization effect on *E. coli in vitro*, indicating that hyperthermia (at the given level) and CDT (at the given agent concentration) cannot compromise the viability of *E. coli*.

Next, we further investigated the residual level of *E. coli* in tumor tissues after *in vivo* treatment by the present agent upon NIR irradiation (T6) on TRIM-infected tumors, compared to Au@Cu_{2-x}Se plus irradiation (T7) and PBS plus irradiation (T2), respectively (Supplementary Fig. 25). Under the given irradiation settings and time, the tumor temperature elevation for T2 was less than 40°C (Supplementary Fig. 46a). The findings demonstrated that, although both delivered significant temperature elevation, T7 had a negligible sterilization effect whereas T6 had a strong sterilization effect. The results confirm that the strong sterilization is primarily attributed to the antibacterial effect of Ag ions rather than hyperthermia.

Following this reviewer's suggestion, we have included the results and associated discussion in the revised manuscript (Pages 8 and 12) and SI as follows:

*“To deconvolute the role of sterilization, we used Au@Cu_{2-x}Se NPS as comparison, as they had excellent PTT and CDT effect [33] and similar morphology as that of Au@Ag₂Se-FA. Despite the significant hyperthermia, Au@Cu_{2-x}Se induced negligible sterilization effect on *E. coli in vitro/vivo* (Supplementary Figs. 16 and 25).”*

*“The residual *E. coli* level in T6 tumor tissues was examined (Supplementary Fig. 25), confirming the potent sterilization of NPs+NIR.”*

“The strong sterilization of NPs+NIR was primarily attributed to the antibacterial effect of Ag ions rather than hyperthermia.”

Supplementary Fig. 25 | **a**, Representative photos for bacteria culture analysis and **b**, bacteria density measured from the breast tumors of T2, T6, T7 after various treatments (*n* = 3).

--- Along the line of deconvoluting the particle/NIR effects, it would be important to include cytokine data from healthy animals that have undergone particle/NIR therapy. This is to ensure that the composite particle is not upregulating or downregulating key immune signals. Pro and anti-inflammatory cytokine analysis from draining lymph nodes would suffice for this purpose.

Answer: We thank the reviewer for raising this constructive comment. Following this reviewer's suggestion, we have evaluated the cytokine data from the healthy mice treated with the therapies. The results are described in the revised manuscript (Page 9) as follows:

"The effects of various therapies on immune cytokines in serum and inguinal lymph nodes of healthy mice were also studied (Supplementary Fig. 33): Au@Ag₂Se-FA or the combination of Au@Ag₂Se-FA and irradiation could not up-regulate or down-regulate the key immune cytokines (IL-6, IFN- γ , IL-17, and IL-10) of healthy mice. In contrast, E. coli injection increased the expression of IL-10 and IL-17 to 108% and 112% in serum, and 130% and 125% in inguinal lymph nodes, respectively, as that of Control."

Supplementary Fig. 33 | Cytokine concentration of IL-6, IFN- γ , IL-17, and IL-10 in **a**, serum and **b**, inguinal lymph nodes of healthy mice treated with i) i.v. injection of Au@Ag₂Se-FA (NPs) ii) i.v. injection of Au@Ag₂Se-FA plus irradiation (NPs+NIR), and iii) intradermally injected with E. coli (EC) (n = 5), with untreated healthy mice as the control. There is no difference between the NPs and NPs+NIR groups compared to Control, indicating that Au@Ag₂Se-FA or the combination of Au@Ag₂Se-FA and irradiation cannot up-regulate or down-regulate the key immune cytokines (IL-6, IFN- γ , IL-17, and IL-10) of healthy mice. Data are presented as mean \pm standard deviation. One-way ANOVA with Tukey's post hoc test. * P < 0.05.

--- Minor comment: Demonstrating the *in vivo* effects in at least two different cancer models will further bolster the generality of the therapeutic approach. In that event and in both cancer models, it needs to be understood from the data that the synergistic drug/immunotherapy and sterilizing effect is not partly or entirely due to additional mechanisms at work such as hyperthermia in the tumor microenvironment.

Answer: We are grateful to this reviewer for providing constructive comments. Following this reviewer's

suggestion, we employed the second cancer model, *i.e.*, the prostate cancer model (RM-1), and evaluated the immunosuppressive effect of antitumor induced by TRIM and the antitumor efficacy of the treatments on the RM-1 tumor. We have included the additional results and associated descriptions in the revised manuscript (Page 11) and SI as follows:

“Generality of the therapeutic approach. *To illustrate the generality of TRIM-induced immunosuppression in the TME and the efficacy of the TRIM-targeting sterilization-antitumor approach, another cancer model, i.e., the prostate cancer model (RM-1), was investigated. The RM-1 tumor-bearing BALB/c mice were divided into four groups (n = 5), with two groups i.t. injected with E. coli (RM-T2 and RM-T4). RM-T2 received no additional treatment. Three days later, NIR irradiation was applied to RM-T3 and RM-T4 12 h after i.v. injection of Au@Ag₂Se-FA. RM-C was the untreated control group. TRIM, like breast cancer, exacerbated immunosuppression in prostate cancer: six days after TRIM infection, intratumoral TNF- α , IL-6, IFN- γ and PD-1 expression in RM-T2 group decreased to 76%, 64%, 64% and 76%, whereas IL-17, TGF- β , and IL-10 increased to 171%, 160% and 149%, respectively, compared to the control group (RM-C) (Fig. 7a). Meanwhile, helper T cells, cytotoxic T cells and M1 TAMs in RM-T2 fell to 45%, 44%, and 85%, respectively, whilst M2 TAMs grew to 180% as compared to RM-C (Figs. 7b and 7c, Supplementary Figs. 42 and 43). Tumors in RM-T2 developed faster than tumors in RM-C. NPs+NIR can effectively reverse the TRIM-induced immunosuppressive effect: the expression of TNF- α , IL-6, IFN- γ , PD-1, and IL-17 in RM-T3 (RM-T4) increased to 168% (165%), 258% (277%), 289% (302%), 511% (534%), and 204 % (211%) compared with RM-C, respectively; TGF- β and IL-10 decreased to 43% (43%), and 53% (59%), respectively; the helper and cytotoxic T cells and M1 TAMs increased to 207% (209%), 250% (251%), and 180% (173%), respectively; M2 TAMs decreased to 51% (50%). Consistently, significant tumor inhibition was attained by NPs+NIR (RM-T3 and RM-T4) (Figs. 7d–f, Supplementary Fig. 44), delivering a high inhibition ratio of ~100%. No prostate tumor recurrence occurred one month after the therapy, demonstrating the universality and long-term efficacy of our therapeutic approach.”*

Fig. 7 | In vivo therapeutic effect on the RM-1 cancer model. a, Intratumoral expression level of TNF- α , IL-6, IL-17, IFN- γ , PD-1, IL-10, and TGF- β upon the various treatments ($n = 5$). **b**, Flow cytometric analysis for the numbers of the helper (CD3⁺, CD4⁺) and cytotoxic (CD3⁺, CD8⁺) T cells, and **c**, M1 (CD86⁺, CD11b⁺) and M2 (CD206⁺, F4/80⁺) TAMs per 100,000 cells from the tumors three days after the treatments ($n = 5$). **d**, Average tumor growth curves, **e**, mean tumor weight, and **f**, body weight of mice in different groups ($n = 5$). Data are presented as mean \pm standard deviation. One-way ANOVA with Tukey's post hoc test (**a-c**, **e**) or two-tailed Student's *t*-test (**d**). * $P < 0.05$, ** $P < 0.01$, *** $P < 0.005$, and **** $P < 0.001$.

Supplementary Fig. 42 | Representative flow cytometric gating strategy for help ($CD3^+$, $CD4^+$) and cytotoxic ($CD3^+$, $CD8^+$) T cells panel in the tumors of the RM-C, RM-T2, RM-T3, and RM-T4 groups three days after the various treatments. The RM-T2 and RM-T4 groups were treated by i.t. injection of *E. coli* three days before the treatments. RM-T2 had no subsequent treatment. RM-T3 and RM-T4 were treated by NIR irradiation 12 h after i.v. injection of $Au@Ag_2Se-FA$. RM-C was the control group without any treatment.

Supplementary Fig. 43 | Representative flow cytometric gating strategy for M1 ($CD86^+$, $CD11b^+$) and M2 ($CD206^+$, $F4/80^+$) TAMs panel in the tumors of the RM-C, RM-T2, RM-T3, and RM-T4 groups three days after the various treatments.

Supplementary Fig. 44 | Digital photos of the mice and tumors (collected from the mice) 24 days after the various treatments ($n = 5$).

Reviewer: #3

Comments:

The work by Wang et al. addressed novel and important research questions on the role of tumor-resident intracellular microbiota (TRIM) in immunosuppressive tumor microenvironment. Further, they have reported a promising single nanotherapeutic formulation, which exhibits photothermal, chemodynamic, chemotherapeutic, and bacteria-killing properties simultaneously upon NIR. The impact and interest of research question deems appropriate for Nature Communications. However, the contribution of TRIM and sterilization in the tumor appears small, and more data are needed to strengthen the story. For example, bacterial infection in mice (without tumors) is needed to tease out anti-bacteria immunity and anti-cancer immunity. Standard antibiotic (ceftriazone) appears to have minimal effect in controlling tumors (e.g., in NS2 group). Further, the difference in efficacy of sterilizing NP

vs. non-sterilizing NP also appears to be small (e.g., T6 vs. T7). Longer-term survival profile of T6 vs T7 will be needed to provide more convincing evidence. Other following issues on premise, study design, data reporting, and statistics also need to be addressed before the reviewer could recommend the article for publication.

Answer: We are very grateful to the reviewer for taking the time to evaluate our manuscript and providing detailed, constructive comments. Following this reviewer’s suggestions, we have performed additional experiments and included the results and discussion in the revised manuscript and SI. Please refer to our point-by-point responses below.

1. Issues on the premise regarding TRIM in cancer and the proposed nanotherapeutic.

a) Intratumoral injection of *E. coli* could trigger a lot of host’s immune response against bacteria itself (especially innate immunity). Some of the observed effects in cytokine levels could be bacteria-related and not tumor-related. Current data do not provide insight on this yet. Data on mouse injected with bacteria alone is needed.

Answer: We thank the reviewer for raising the constructive comment. Following the reviewer’s suggestion, we have carried out additional experiments for the tumor-free mouse treated by the bacteria injection and further confirmed that injection of *E. coli* can alter the expression of specific immune cytokines (e.g., IL-10 and IL-17) [Cancer Discov. 8, 403–416 (2018)]. We have included the results in Supplementary Fig. 23 and added the associated descriptions in the revised manuscript (Page 9) as follows:

“The effects of various therapies on immune cytokines in serum and inguinal lymph nodes of healthy mice were also studied (Supplementary Fig. 33): Au@Ag₂Se-FA or the combination of Au@Ag₂Se-FA and irradiation could not up-regulate or down-regulate the key immune cytokines (IL-6, IFN-γ, IL-17, and IL-10) of healthy mice. In contrast, *E. coli* injection increased the expression of IL-10 and IL-17 to 108% and 112% in serum, and 130% and 125% in inguinal lymph nodes, respectively, as that of Control.”

Supplementary Fig. 33 | Cytokine concentration of IL-6, IFN-γ, IL-17, and IL-10 in **a**, serum and **b**, inguinal lymph nodes of healthy mice treated with i) i.v. injection of Au@Ag₂Se-FA (NPs) ii) i.v. injection of Au@Ag₂Se-FA plus irradiation (NPs+NIR), and iii) intradermally

injected with *E. coli* (EC) ($n = 5$), with untreated healthy mice as the control. There is no difference between the NPs and NPs+NIR groups compared to Control, indicating that Au@Ag₂Se-FA or the combination of Au@Ag₂Se-FA and irradiation cannot up-regulate or down-regulate the key immune cytokines (IL-6, IFN- γ , IL-17, and IL-10) of healthy mice. Data are presented as mean \pm standard deviation. One-way ANOVA with Tukey's post hoc test. * $P < 0.05$.

b) Injecting bacteria to tumor could promote ulceration, and tumor volume alone is not a good measure of tumor burden. Additional analysis of tumor burden is needed to substantiate this.

Answer: We thank the reviewer for raising the constructive comment. Following the reviewer's suggestion, we have added the additional analysis of tumor burden (i.e., mean tumor weight) when discussing tumor growth in the revised manuscript (Page 4) as follows:

"The tumor growth was monitored : while the tumor volume in the control group enlarged to 204.1% at Day 16, the tumors in the EC group already increased to 215.6% at Day 4 and to 380.6% at Day 16 (Fig. 2a); the mean tumor weight in the EC group was consistently higher than that in Control (Supplementary Fig. 2)."

Supplementary Fig. 2 | a, Tumor volume ($n = 4$) and **b**, mean tumor weight ($n = 3$) of *E. coli*-infected group, compared with the non-infected mice group (Control). Data are presented as mean \pm standard deviation. One-way ANOVA with Tukey's post hoc test (**b**) or two-tailed Student's *t*-test (**a**). * $P < 0.05$.

c) Justification of 500,000 *E. coli* cells used in the tumor model is needed. How representative is this to real patient scenario?

Answer: We thank the reviewer for the important suggestion. "5 \times 10⁵ *E. coli*" was a typo. In our work, we used 1 \times 10⁵ CFU. In the reported bacterial infection models, injection of 1 \times 10⁵~1 \times 10⁶ CFU of bacteria cells is often employed to represent the real-life infection [Science, 357, 1156–1160 (2017); Cell, 185, 1356–1372 (2022); Nat. Biomed. Eng., 6, 32-43 (2022)]. Following this reviewer's suggestion, we have added the following justification in the revised manuscript (Page 4) as follows:

"To construct the TRIM model, 1 \times 10⁵ CFU *E. coli* cells were intratumorally (i.t.) injected into 4T1 breast tumor-bearing mice (EC), with the mice i.t. injected with sterile phosphate buffer solution (PBS) as Control. The number of bacteria was chosen to be representative of the normal infection [6,12,29]."

[6] Geller, L. T. et al. Potential role of intratumor bacteria in mediating tumor resistance to the chemotherapeutic drug gemcitabine.

Science **357**, 1156–1160 (2017).

[12] Pushalkar, S. et al. The pancreatic cancer microbiome promotes oncogenesis by induction of innate and adaptive immune suppression. *Cancer Discov.* **8**, 403–416 (2018).

[29] Zheng, D. et al. Biomaterial-mediated modulation of oral microbiota synergizes with PD-1 blockade in mice with oral squamous cell carcinoma. *Nat. Biomed. Eng.* **6**, 32–43 (2022).

d) The authors assessed impact of TRIM on tumor's immune milieu by IHC analysis of CD3, CD86, and ARG. The authors claimed CD86 to represent M1 macrophages and ARG to represent M2 macrophages. This is not correct because CD86 is also expressed in DC, monocytes, and T cells. ARG (presumable ARG-1 – please clarify) is also expressed in other myeloid cells (e.g., MDSC).

Answer: We appreciate the reviewer raising this point. Following this reviewer's suggestion, we have employed flow cytometry to further analyze the number of M1-TAMs (CD86⁺, CD11b⁺) in addition to IHC analysis. M2 TAMs were evaluated by ARG-1, known as a classic M2 marker [*Nat. Immunol.*, **9**, 1399–1406 (2008); *Cell Rep.*, **17**, 684–696 (2016); *Cell Metab.*, **34**, 487–501(2022)]. We have included the additional experimental results in Supplementary Fig. 4 and provided the associated descriptions in the revised manuscript (Page 5) as follows:

“Furthermore, as revealed by the immunohistochemistry (IHC) analysis (Fig. 2f and Supplementary Fig. 3), the numbers of T cells (CD3⁺) and M2 TAMs (ARG-1⁺) evidently varied after injection: at Day 15, the number of T cells reduced to 27%, while M2 TAMs increased to 286%. In contrast, these cell numbers of the control group were barely varied. Consistently, flow cytometric analysis (Supplementary Fig. 4) demonstrated that *E. coli* injection reduced the number of M1 TAMs (CD11b⁺, CD86⁺) to 65%.”

Supplementary Fig. 4 | a, Flow cytometric gating strategy for M1 TAMs (CD11b⁺, CD86⁺) panel of Control and EC. **b**, The number of CD11b⁺ and CD86⁺ M1 TAMs per 50,000 cells of tumors in Control and EC (n = 4). Data are presented as mean ± standard deviation. One-way ANOVA with Tukey's post hoc test. ** P < 0.01.

--- The flow analysis that the author later did is more appropriate. However, that presented dataset is not complete (only T2, T6 and T7 data are shown in Fig. S25). The authors should present all flow data for T1-T7. Flow cytometry of T1 vs. T2 will be more convincing in showing the role of TRIM in tumor's immune milieu.

Answer: We thank the reviewer for the important suggestion. The T1–7 groups for the *in vivo* experiments corresponded to the mice treated by i) PBS plus NIR irradiation (T1 and T2), ii) Au@Ag₂Se-FA only without irradiation (T3 and T4), iii) Au@Ag₂Se-FA (the agent used in this work) plus irradiation (T5 and T6), and iv) Au@Cu_{2-x}Se (the agent used as a control group) plus irradiation (T7), where the mice of T2, T4, T6, and T7 were *i.t.* injected with *E. coli*. Following this reviewer’s suggestion, we have included the flow cytometric analysis for T1–T7 (without T3–5) in Supplementary Fig. 26. In fact, the results of the intratumoral cytokines for T5 were already presented in Fig. 5d. The reasons that we did not include T3 and T4 are as follows: i) the tumor growth rates of T3 and T4 groups were very similar to those of T1 and T2, respectively (Fig. 5b); ii) Au@Ag₂Se-FA alone (T3 and T4) cannot up-regulate or down-regulate the key immune cytokines (Supplementary Fig. 33); iii) considering the Ethics Committee for Animal Experimentation guidelines, we minimized the groups and mice as much as we could. Following this reviewer’s suggestion, we have compared the key immune cytokines and flow cytometric results between T1 and T2 in the revised manuscript (Page 8) as follows:

“The T2 group showed a stronger immunosuppression effect than T1: the helper T cells (CD3⁺, CD4⁺), cytotoxic T cells (CD3⁺, CD8⁺), and M1 TAMs (CD86⁺, CD11b⁺) in T2 decreased to 59%, 27%, and 60%, respectively, while M2 TAMs (CD206⁺, F4/80⁺) increased to 141% as that of T1 (Supplementary Fig. 26). The variation between T1 and T2 further confirmed the aggravated antitumor immunosuppression effect of TRIM in breast cancer.”

Supplementary Fig. 26 | a, Flow cytometric analysis of the helper (CD3⁺, CD4⁺) T cells, **b**, cytotoxic (CD3⁺, CD8⁺) T cells, **c**, M1 (CD86⁺, CD11b⁺) and **d**, M2 (CD206⁺, F4/80⁺) TAMs in the TME (n = 5), and **e**, quantification of these cells after the various treatments, including i) PBS plus NIR irradiation (T1 and T2), ii) Au@Ag₂Se-FA plus irradiation (T6), iii) Au@Cu_{2-x}Se plus irradiation (T7), where the mice of T2, T6, and T7 were *i.t.* injected with *E. coli*. Data are presented as mean ± standard deviation. One-way ANOVA with Tukey’s post hoc test *P < 0.05, **P < 0.01, ***P < 0.005 and ****P < 0.001.

e) Metastasis aspect of the disease was not discussed or studied. NIR is limited to superficial solid tumors. It is

important to understand if this treatment could sufficiently curb metastasis diseases or this technology will only benefit localized disease. For example, the triggered immunity by NPs+NIR could potentially affect distal or metastatic tumors (which will not be irradiated). Additional study showing abscopal effect of NPs+NIR will substantially enhance the impact of the work.

Answer: We highly appreciate the reviewer's constructive comments. Following this reviewer's suggestion, we have established a dual tumor model to address the abscopal effect of NPs+NIR. The results have been included in Supplementary Fig. 30 and described in the revised manuscript (Pages 9 and 16) as follows:

***“Effect on the distal tumor.** To explore the systemic immunological effects on the distal tumor (mimicking tumor migration), we established a dual-tumor model and only treated the primary tumor with i.t. injection of Au@Ag₂Se-FA NPs plus NIR, leaving the distal tumor untreated (Supplementary Fig. 31). Remarkably, effective systemic antitumor immune responses were demonstrated in the distal tumor: nine days post-treatment, the expression of TNF- α , IL-6, IFN- γ , PD-1, and IL-17 in the distal tumor increased to 205%, 245%, 192%, 514%, and 205%, while IL-10 and TGF- β decreased to 41% and 34%, respectively, compared with the distal tumor of the control group (the primary tumor was i.t. injected with PBS and then irradiated). Twenty-one days post-treatment, the average distal tumor weight was only 30.2% that of Control, while the primary tumor was completely eliminated, showing no recurrence. These findings suggest that such sterilization-combined antitumor therapy can trigger pronounced systemic antitumor immune responses, potentially addressing malignant tumor migration.”*

“To construct the dual-tumor model, 1.0×10^6 4T1 cells were subcutaneously injected into the right back of each mouse to establish the primary tumor, and 5.0×10^5 4T1 cells were subcutaneously injected into the left back of each mouse to obtain the distal tumor. When the tumors grew to ~8 mm in diameter, the tumor-bearing mice were randomly divided into different groups for various treatments.”

Supplementary Fig. 31 | In vivo therapeutic and immune effect on the distal tumor. Four different tumor groups are investigated, including i) the primary tumors treated with i.t. injection of Au@Ag₂Se-FA (50 μL, 3 mg·mL⁻¹) plus irradiation for 10 min (808 nm, 0.36 W·cm⁻²) (P-T) and ii) their corresponding distal tumors (D-T), and iii) the primary tumors treated with i.t. injection of PBS (50 μL) plus irradiation for 10 min (808 nm, 0.36 W·cm⁻²) (P-Control) and iv) their corresponding distal tumors (D-Control). Note that the distal tumors receive no treatment. **a**, In vivo IRT images of 4T1 tumor-bearing mice showing that the P-T group tumor reaches ~62 °C, while the C-T group tumor temperature remains unchanged upon the 10 min irradiation. **b**, Intratumoral expression of TNF-α, IL-18, IFN-γ, PD-1, IL-17, TGF-β, and IL-10 for the D-T and D-Control groups (n = 5). **c**, Tumor growth curves of the P-T, C-T, P-Control and C-Control groups (n = 5). **d**, Digital photos, and **e**, mean weight of the primary and distal tumors collected 21 days after the various treatments (n = 5). Data are presented as mean ± standard deviation. One-way ANOVA with Tukey's post hoc test (**b**, **e**). Two-tailed Student's t-test (**c**). *P < 0.05, **P < 0.01, ***P < 0.005, and ****P < 0.001.

f) Does low efficacy of ceftriazone alone (NS2) in Fig.6 and Fig. S27 suggest small contribution of sterilization to tumor therapy because the author mentioned ceftriazone as being a powerful antibiotic in clinic already? Lobaplatin (used in NS3 and NS4 group) has cancer-killing effect (shown in Fig. S26), so it does not show sterilization effect alone. Please address/discuss this finding more in Discussion section.

Answer: We thank the reviewer for raising this point. In Fig. 6d and Supplementary Fig. 27 (Supplementary Fig. 39 in the revised SI), ceftriazone alone (NS2) can reduce immunosuppression as compared with the untreated group (NS1), showing increased TNF-α level and help T cells and decreased TGF-β level and M2-TAMs compared with those of NS1. The mean tumor weight and tumor growth rate (Figs. 6a and 6b) by applying ceftriazone alone also showed slower tumor growth compared with the untreated group. However, as ceftriazone has only an antibacterial effect without noticeable antitumor function, its antitumor efficacy was reasonably weak. Still, these

findings suggest that the powerful antibiotic alone can already ease immunosuppression and retard tumor growth to some extent.

Following this reviewer's suggestion, we have included additional experimental results together with the associated discussion in the revised manuscript (Page 10) and SI as follows:

"Tumors in NS2 showed a lower growth rate than in NS1 (with a tumor weight of 75.7% as that in NS1 on Day 16) (Figs. 6a and 6b). Combining the increased TNF- α level and help T cells and decreased TGF- β level and M2-TAMs of NS2 as compared with NS1 (Figs. 6c and 6d, Supplementary Fig. 39), these findings suggest that the antibiotic alone can already alleviate immunosuppression and retard tumor growth to some extent."

"NS3 showed even slower tumor growth than NS2, with a tumor volume of 126.7% on Day 16 compared to that on Day 0. NS4 delivered an inhibition rate of 77.4% 16 days post-treatment. NS5 provided a tumor inhibition rate of 100%. All these results emphasized the importance of combining sterilization with antitumor therapy for immune activation, not limited to TRIM-infected tumors."

g) The impact on T cells was discussed and reported. However, it is unclear if the generated T cells is tumor-specific or bacteria-specific. Evidence of tumor-specific immune response could be provided by either tetramer staining of splenocytes, or an ex vivo stimulation assay of splenocytes with tumor lysate or relevant tumor antigens, or an in vivo tumor rechallenge study (where tumors are implanted to the cured mice months after treatment).

Answer: We appreciate the reviewer's constructive advice. Following this reviewer's suggestion, we have carried out a tumor rechallenge test *in vivo*, which demonstrates that the generated T cells are tumor-specific. The results demonstrate strong systemic antitumor immune responses and long-term immunological memory against tumor relapse, enabled by Au@Ag₂Se-FA NPs plus NIR irradiation. We have included the results and associated descriptions in the revised manuscript (Page 9) and SI as follows:

"To demonstrate the long-term durability of the immune responses, 45 days after the treatment, mice of T5 and T6 (n = 9) that had survived from the first inoculation of 4T1 tumors were re-challenged via subcutaneous administration of 4T1 tumor cells (1.0×10^6) at the previously tumor-bearing area without any additional treatments (R-T5 and R-T6 groups). 10 days after the re-challenge, the mice's blood and spleens were evaluated by flow cytometric analysis (Fig. 5f, Supplementary Figs. 27–29). Excitingly, all these mice still exhibited significant immune responses. The number of NK cells (CD3⁻, CD49b⁺) [42] in R-T5 and R-T6 grew to 455% and 481%, the $\gamma\delta$ T cells (CD3⁺, TCR β ⁻) [43] increased to 363% and 389%, and the memory T cells (CD8⁺, CD44^{high}, CD62L^{low}) [22] increased to 601% and 606%, as compared to the untreated tumor-bearing mice (Control). Both R-T5 and R-T6 showed no sign of tumors 21 days after the re-challenge (Supplementary Fig. 30). The results emphasized the long-lasting immunological memory against 4T1 malignancy by NPs+NIR treatment. Such memory may have played a crucial role in the complete cure."

Fig. 5 | Therapeutic effect of Au@Ag₂Se-FA *in vivo*. *f*, Flow cytometric analysis for the number of NK cells (CD3⁻, CD49b⁺), $\gamma\delta$ T cells (CD3⁺, TCR β ⁻), and memory T cells (CD8⁺, CD44^{high}, CD62L^{low}) per 100,000 cells in blood or spleen of the R-T5 and R-T6 groups, with the results from the untreated 4T1 tumor-bearing mice as Control (*n* = 5).

Supplementary Fig. 27 | Representative flow cytometric gating strategy for NK cells (CD3⁻, CD49b⁺) panel in blood of the R-T5, R-T6, and control groups.

Supplementary Fig. 28 | Representative flow cytometric gating strategy for $\gamma\delta$ T cells (CD3⁺, TCR β ⁻) panel in blood of the R-T5, R-T6, and control groups.

Supplementary Fig. 29 | Representative flow cytometric gating strategy for memory T cells (CD8⁺, CD44^{high}, CD62L^{low}) panel in spleens of the R-T5, R-T6, and control groups.

Supplementary Fig. 30 | a, Digital photo (tumors in Contol at Day 21) and **b**, mean weight of the tumors collected from the mice 21 days after the various treatments ($n = 4$).

h) What happens to the released Au, Ag, and Se in the body? Short-term study reveals no effects in livers and kidneys. However, long-term effect or deposition of these elements to organs is not described. It is unclear how many % of what's injected can be recovered in urine cumulatively (e.g., XX% of administered Au, Ag, and Se is recovered in urine by YY days). Please perform mass balance analysis (with data from Fig. S22) to address this question.

Answer: We appreciate the reviewer's constructive comments. Following this reviewer's suggestion, we have carried out additional experiments and included the results in Supplementary Fig. 22 and added the associated descriptions in the revised manuscript (Pages 7 and 8) as follows:

"The urinary excretion of Au@Ag₂Se-FA was also investigated (Supplementary Fig. 22): ~15% and 11% of Au and Ag were excreted in urine five days after injection, and ~50% of Se was excreted in urine three days after injection. The high concentration of Se, Ag, and Au ions detected in urine suggested that Au@Ag₂Se-FA NPs could be metabolized via the kidney and excreted out of the body in the urine. Combined with the biodistribution of Au@Ag₂Se-FA in the mononuclear phagocyte system organs, the vital role of nephritic and hepatic clearance in the excretion of degradation products was confirmed."

Supplementary Fig. 22 | Concentration of a, Se ions, **b**, Ag ions, and **c**, Au ions in urinary at different time points after i.v. injection of Au@Ag₂Se-FA dispersion ($n = 3$). Data are presented as mean \pm standard deviation. After injection, Balb/c mice were housed in metabolic cages. 4 h before each time point, their urine was collected, then digested using aqua regia, and measured using ICP-MS.

i) Treg, which is part of CD3+CD4+, is not measured or addressed. CD8/Treg is another important indicator beyond

CD8 T cells alone.

Answer: We thank the reviewer for raising this important point. According to the literature [*Nat. Biomed. Eng.* 2020, 4, 1102–1116; *Nat. Biomed. Eng.*, 5, 1377-1388 (2021)], cytotoxic (CD3⁺, CD8⁺) T cells and M1-TAMs (CD86⁺, CD11b⁺) are the major/sufficient indicators for antitumor immune regulation. As we have stated in the manuscript, “The correlation between TRIM-targeting sterilization and immune regulation is more profound and complicated than what have been presented in this work, deserving further investigation.” Although we attempted to be as thorough as possible, it is not practical to include all the results and details in a single article. This is a fascinating area that is still in a very early stage. Following this reviewer’s suggestion, we will delicately explore the antitumor immune regulation of TRIM to the various immune cells in our future work, including Treg and CD8/Treg.

2. Issues on material characterization and justification of folic acid

j) The authors did a great job on establishing ‘identity’ of the synthesized material. However, composition analysis is not adequate yet. For example, how much of folic acid got loaded on the nanoparticle and how did the authors measure it? Also, in the method section, there appears to be no purification after folic acid loading. Please clarify and elaborate.

Answer: We appreciate the reviewer offering the important advice. Following the reviewer’s suggestions, we have evaluated the FA load by UV-vis-NIR absorption spectra (Supplementary Fig. 5c and 5d) and included the NPs’ purification procedure after folic acid loading in ‘Methods’ in the revised manuscript (Page 14) as follows:

Supplementary Fig. 5 | a, Surface potential, **b**, Fourier transform infrared spectra, and **c**, UV-vis-NIR spectra of Au@Ag₂Se before (bare NPs, NPs-B) and after reaction with folic acid-polyethylene glycol-thiol (FA-PEG-SH) (Au@Ag₂Se-FA). **d**, Absorbance of FA-PEG-SH dispersions at 282 nm and various concentrations. According to **c** and **d**, the FA-PEG-SH load on 100 µg Au@Ag₂Se-FA is estimated to be ~29 µg (~3.8 µg of FA).

“The as-prepared NPs (3 mg) were dispersed in FA-PEG-SH solution (100 µg·mL⁻¹, 50 mL) and kept at 4 °C upon

vigorous stirring for 24 h to obtain the coated NPs ($\text{Au@Ag}_2\text{Se-FA}$). The NPs were finally centrifuged and washed three times with DI water for further experiments.”

--- DLS data (hydrodynamic size) of NP is currently presented in the supplementary data. However, the size of NP in solution and biological matrix (e.g., serum) is important for nanomedicine. The authors should thus at least mention Z-average and PDI in the main paper.

Answer: We appreciate the reviewer raising this important point. Following the reviewer’s suggestion, we have evaluated the hydrodynamic size of $\text{Au@Ag}_2\text{Se-FA}$ NPs dispersed in DI water, PBS, DMEM, and 1640, by dynamic light scattering, respectively, and addressed the Z-average and PDI in the revised manuscript (Page 5) as follows:

“Dynamic light scattering (DLS) was used to determine the hydrodynamic size in various physiological solutions (Supplementary Fig. 6), revealing a Z-average of 49.4 nm and a polydispersity index of 0.03.”

Supplementary Fig. 6 | a, Digital photos of $\text{Au@Ag}_2\text{Se-FA}$ dispersed in various physiological solutions, including phosphate buffer saline (PBS), Dulbecco’s modified Eagle’s medium with 10% FBS (DMEM), RPMI medium 1640 basic with 10% FBS (1640), and DI water after storing at room temperature for seven days. **b,** Hydrodynamic size of $\text{Au@Ag}_2\text{Se-FA}$ NPs dispersed in DI water, PBS, DMEM and 1640, and **c,** $\text{Au@Ag}_2\text{Se-FA}$ dispersed in DI water stored at room temperature for varied durations.

k) Please confirm that ALL studies in the paper that use NPs (e.g., characterization, stability, Ag+ release) employed the version with folic acid. If no, please provide distinguishing terms between NP with vs. without folic acid. If yes, please include one sentence confirming this in the paper (e.g., “ $\text{Au@Ag}_2\text{Se}$ (which includes folic acid) is used in all studies”).

Answer: We thank the reviewer for raising this important suggestion. Following this reviewer’s suggestion, we have renamed the NPs as $\text{Au@Ag}_2\text{Se-FA}$ to emphasize their coating with folic acid and included one sentence clarifying this point in the revised manuscript (Page 5) as follows:

“To enable tumor targeting, FA was equipped on the NPs’ surface (Fig. 1 and Supplementary Fig. 5), and the resultant $\text{Au@Ag}_2\text{Se-FA}$ NPs were used in all studies.”

l) Has the importance and necessity of folic acid been experimentally shown and proven for this specific nanoformulation yet (e.g., data of NP with vs. without folic acid)? Please discuss and include relevant data.

Answer: We thank the reviewer for raising this advice. In fact, it has been well demonstrated in the literature that surface modification with folic acid (FA) can promote the tumor-targeting function of nanoagents; FA has been commonly used for this purpose [*Nanoscale*, 10, 8536 (2018); *Nano Research*, 13: 1389–1398 (2020)]. Following this reviewer's suggestion, we have carried out additional experiments to compare the tumor-targeting accumulation of Au@Ag₂Se-FA with that of bare Au@Ag₂Se *in vivo* using IRT imaging. The results have been included in Supplementary Fig. 21 in the revised SI and addressed in the revised manuscript (Page 7) as follows:

“The agent's tumor-targeting accumulation was also verified by the substantial tumor-selective photothermal effect after i.v. injection of Au@Ag₂Se-FA (Supplementary Fig. 21), where the surface modification of FA promoted the tumor-targeting effect.”

Supplementary Fig. 21 | *In vivo* IRT images of 4T1 tumor-bearing mice. Whilst the tumor temperature of mice treated by i.v. injection of Au@Ag₂Se is < 46 °C upon 10 min NIR irradiation (808 nm, 0.36 W·cm⁻²), the tumor temperature of mice treated by i.v. injection of Au@Ag₂Se-FA quickly increases to ~55 °C upon the same irradiation, suggesting the enhanced tumor-targeting effect by FA modification.

3. Issues on study designs (missing controls), data presentation, and data interpretation

m) Immune response *in vitro* needs a PI-Au@Ag₂Se control because that would rule out hyperthermia effect in the observed phenotype. The justification/relevance of M0 macrophage model is not described. THP-1 cell was not mentioned outside method section.

Answer: We thank the reviewer for raising the constructive comment. Following the reviewer's suggestion, we have carried out additional experiments on M0 macrophages treated by PI-Au@Ag₂Se-FA and included the results and associated descriptions together with the justification of the M0 macrophage model in the revised manuscript (Page 7) and SI as follows:

“Human leukemia monocyte THP-1 cells were treated with 12-O-tetradecanoylphorbol-13-acetate to get M0 macrophages [39]. After being treated with i) Au@Ag₂Se-FA alone (G1), ii) PI-Au@Ag₂Se-FA alone (G2), iii) NIR

irradiation only (G3), and iv) Au@Ag₂Se-FA plus NIR irradiation (G4), all groups showed high cell viability (> 93.7%), suggesting rather low/no cytotoxicity to M0 macrophages (Supplementary Fig. 17)."

Supplementary Fig. 17 | Cell viability of M0 macrophages after treated with i) pristine Au@Ag₂Se-FA (100 μg·mL⁻¹) (G1), ii) PI-Au@Ag₂Se-FA (100 μg·mL⁻¹) (G2), iii) NIR irradiation only (0.8 W·cm⁻², 10 min) (G3), and iv) Au@Ag₂Se-FA plus NIR irradiation (G4), with the untreated M0 macrophage cells as Control.

n) Fig. S12 shows a much more pronounced effect of PI-Au@Ag₂Se over Au@Ag₂Se+NIR. Please discuss. Fig. S16 doesn't have NPs+NIR, which is a key group.

Answer: We thank the reviewer for raising the comments. The green fluorescence in Supplementary Fig. 12 (Supplementary Fig. 13 in the revised SI) indicated the ROS generation in the cells. As the PI-Au@Ag₂Se-FA had a much lower inhibition capability than NPs+NIR, the survival rate (reflected by the density of the cells) of cells treated by the former was significantly higher than the latter; in contrast, only a small number of cells were observed for the latter due to the high efficacy. The signal intensity of ROS in each remaining cell of NPs+NIR was nearly identical to that in the cells treated by PI-Au@Ag₂Se-FA.

Following the reviewer's suggestion, we have carried out additional experiments and included the results of NRs+NIR in Supplementary Fig. 16 (Supplementary Fig. 18 in the revised SI) and described in the revised manuscript (Page 7) as follows:

"Reactive nitrogen species (RNS) and ROS were largely promoted in M0 macrophages of G2 and G4, with RNS being 287% and 284% and ROS being 202% and 195% as that of Control, respectively (Supplementary Fig. 18)."

Supplementary Fig. 18 | a, Nitric oxide (NO) and b, ROS release from M0 macrophages by i) Au@Ag₂Se-FA (100 μg·mL⁻¹) without

irradiation (G1), ii) PI-Au@Ag₂Se-FA (100 μg·mL⁻¹) (G2), iii) NIR irradiation only (0.8 W·cm⁻², 10 min) (G3), and iv) Au@Ag₂Se-FA plus NIR irradiation (G4) (n = 5). Data are presented as mean ± standard deviation. One-way ANOVA with Tukey's post hoc test. **** P < 0.001.

o) Fig. S21b should be of the same format as Fig. S21a to maintain coherence. In particular, x-axes should be biomarkers consistently (e.g., AST, ALP, ALT, CREA, UA, BUN).

Answer: We thank the reviewer for raising this point. Following this reviewer's suggestion, we have modified the format of the figure to:

Supplementary Fig. 36 | Serum biochemistry analysis of the mice treated by NPs+NIR (T6) for **a**, liver function markers including alkaline phosphatase (ALP), aspartate aminotransferase (AST), and alanine aminotransferase (ALT), and **b**, renal function markers including creatinine (CREA, μmol·mL⁻¹), uric acid (UA, μmol·mL⁻¹), and blood urea nitrogen (BUN, mmol·mL⁻¹) (n = 3). The values of liver function marker and renal function markers of T6 are close to those of healthy mice (Normal).

p) Fig. S14 [100 ug/ml Au@Ag₂Se] shows less kill than the same data in Fig. 4b. Please explain why? Clearer and more detailed method section throughout the paper (see section 4) may clarify.

Answer: We thank the reviewer for bringing up this question. The treatments in Supplementary Fig. 14 (Supplementary Fig. 15 in the revised SI) and Fig. 4b were not the same: in the former, 4T1 cells were treated by pre-irradiated Au@Ag₂Se-FA without additional irradiation; for the latter, 4T1 cells were treated by pristine Au@Ag₂Se-FA plus NIR irradiation. In this way, the former only relied on CDT and chemotherapy, while the latter enabled hyperthermia by photothermal conversion of Au@Ag₂Se-FA in addition to CDT and chemotherapy.

q) Efficacy of T6 vs T7 is small, but appears significant. Long-term survival of T7 (in the same manner as Fig. 5C) should be provided to show clear benefits over T6.

Answer: We appreciate the reviewer's comments. As there are quite a number of parallel groups in the present work, it would be easy to become confused. The T6 group was treated with the intended agent, *i.e.*, Au@Ag₂Se-FA, plus NIR irradiation, whereas the T7 group was treated with the control agent, *i.e.*, Au@Cu_{2-x}Se, plus irradiation. Combining the sterilization effect with the antitumor therapies, T7 delivered a much lower efficacy than T6, primarily attributed to its negligible sterilization effect. To further clarify this point, following this reviewer's

suggestion, we have evaluated the life span of the T7 group and included the additional results and associated descriptions in the revised manuscript (Page 12) and SI as follows:

“For the tumor inhibition efficacy, Au@Cu_{2-x}Se plus NIR irradiation produced a tumor inhibition rate of 83.8% on the TRIM-infected tumors (T7) at 12 days post-treatment, which was lower than that on the non-infected tumor (95.7% [33]). NPs+NIR showed close inhibition rate for the tumors with/without TRIM infection (100% vs. 98.9%) (Fig. 5b). Moreover, evident tumor recurrence can already be observed within 16 days after the Au@Cu_{2-x}Se+NIR treatment. Within 58 days post-treatment, every mouse in T7 passed away (Supplementary Fig. 46). In stark contrast, T6 obtained a relapse-free survival of > 700 days (Fig. 5c). All these results further confirmed that combination with sterilization played a crucial role for the immune regulation in TME.”

Supplementary Fig. 46 *a*, In vivo IRT images of 4T1 tumor-bearing mice: whilst the tumor temperature of T2 (treated with PBS injection) is < 40 °C upon 10 min NIR irradiation (808 nm, 0.36 W·cm⁻²), the tumor temperature of T7 (treated with i.t. injection of Au@Cu_{2-x}Se NPs) quickly climbed to ~55 °C upon the same irradiation. *b*, Tumor growth rate in the different groups (n = 5): all groups receive an i.t. injection of *E. coli* followed by one of three treatments: i) i.v. injection of PBS plus NIR irradiation (T2), ii) i.v. injection of Au@Ag₂Se-FA plus NIR irradiation (T6), or iii) i.t. injection of Au@Cu_{2-x}Se plus NIR irradiation (T7). The irradiation conditions are 808 nm, 0.36 W·cm⁻² and 10 min. *c*, Digital photos (tumors in T7 on Day 16 post-treatment) and mean tumor weight from the mice at 16 days after the various treatments (n = 5). *d*, Body weight of mice in the different groups after the various treatments (n = 5). *e*, Survival curves after the various treatments (n = 5). All mice in T7 are dead within 58 days after treatment. Data are presented as mean ± standard deviation.

r) ‘mRNA level’ should be spelled out, instead of ‘gene expression level’.

Answer: We have made the revision as suggested by this reviewer in the revised manuscript as follows:

“Fig. 4 | Therapeutic efficacy and immune response induced by Au@Ag₂Se-FA in vitro. The relative expression of *e*, TNF α , *f*, Arg-1, and *g*, Fizz1 from M0 macrophages treated with i) pristine Au@Ag₂Se-FA (100 μ g·mL⁻¹) (G1), ii) PI-Au@Ag₂Se-FA (100 μ g·mL⁻¹) (G2), iii) NIR irradiation only (0.8 W·cm⁻², 10 min) (G3), and iv) pristine Au@Ag₂Se-FA plus NIR irradiation (G4) (n = 5). Data are presented as mean ± standard deviation. One-way ANOVA with Tukey’s post hoc test ** P < 0.01, **** P < 0.001.”

s) Explanation of Fig. S25 in text is not complete yet. The authors discussed T cell populations for Au@Ag₂Se and then jumped to discuss macrophage populations for Au@Cu_{2-x}Se. Please revise. In another sentence on NS1's cytokine measurement, Fig. 5d seems incorrectly introduced. Please fix or clarify its inclusion as needed.

Answer: Following the reviewer's suggestion, we have made the following revisions to the revised manuscript (Pages 10 and 12):

"We then compared the immune response activated by Au@Ag₂Se-FA to that by Au@Cu_{2-x}Se upon NIR irradiation (Supplementary Fig. 45): although T7 also delivered largely up-regulated expression of intratumoral TNF- α (207%), IL-1 β (140%), IFN- γ (215%) and PD-1 (162%) and down-regulated expression of IL-10 (46%) and TGF- β (55%) as that of Control (T2), the regulation extent of these markers in T7 was much weaker than that in T6, suggesting a more efficient restoration of immune surveillance by Au@Ag₂Se-FA; moreover, the helper T cells (CD3⁺, CD4⁺), cytotoxic T cells (CD3⁺, CD8⁺) and M1 TAMs (CD86⁺, CD11b⁺) in T6 increased to 192%, 282% and 252%, whilst M2 TAMs (CD206⁺, F4/80⁺) fell to 58% (Supplementary Fig. 26)."

"Based on the level of intratumoral TNF- α , IL-1 β , IFN- γ , PD-1, IL-17, TGF- β , and IL-10, NS1 demonstrated aggravated immunosuppression and tumor overgrowth (125% at Day 16) as compared with Control (tumor-bearing mice raised in the sterile environment, T1 in Fig. 5d) (Supplementary Figs. 39 and 40),..."

t) Please consider having a separate session on non-sterilizing nanoparticle, such that 'Discussion' section does not contain a lot of newly introduced data.

Answer: We highly appreciate the reviewer offering this important advice. We have relocated the session discussing therapeutic efficacy and immune responses of the mice raised in a non-sterile environment into the 'Results' section, as suggested by this reviewer.

4. Issues on incomplete method descriptions

u) It is unclear how the authors monitor bacteria burden in the tumor. Please explain more. The reviewer couldn't find such info in method or supplementary section yet.

Answer: We thank the reviewer for raising this advice. Following the reviewer's suggestion, we have added the details in the 'Methods' section of the revised manuscript (Page 17) as follows:

"The bacteria burden of tumor was analyzed as follows: each tumor (0.1 g) was first mashed into tissue homogenate with sterile PBS (5 mL); diluted tissue homogenate (100 μ L) was then uniformly covered on an agar plate and incubated at 37 $^{\circ}$ C for 24 h."

v) Please provide information on all antibody clones, conjugated dyes, vendors, dilution factor, used in all assays.

Answer: We thank the reviewer for raising this advice. Following the reviewer's suggestion, we have added the details in the 'Methods' section of the revised manuscript (Pages 13 and 17) as follows:

"RPM1 Medium 1640 basic (1X), Dulbecco's modified Eagle's medium (1X), PBS (pH = 7.4, 1X), and fetal bovine serum (FBS) were purchased from Thermo Fisher Biochemical Products (Beijing) Co., Ltd."

"All antibodies were purchased from BD Bioscience and used following the manufacturer's instructions."

"The tumor slices were deparaffinized, rehydrated, and then incubated with primary anti-CD3 (Ablonal), anti-CD86 (Ablonal), anti-ARG-1 (Ablonal), and secondary antibodies, followed by staining with DAB and hematoxylin."

w) Method section does not describe sample preparation and staining protocols for IHC (unlike flow cytometry). Further, it is unclear how many images and their field of view (e.g., how many mm³) were being analyzed for IHC per group. Some representative images in the supplementary info do not agree with certain time points in Fig. 2.

Answer: We thank the reviewer for providing this important suggestion. Following the reviewer's suggestion, we have added the details in the 'Methods' section of the revised manuscript (Page 17) as follows:

"To investigate the influences of TRIM on breast tumors and immune regulation in TME, the tumor-bearing mice (tumor diameter of ~8 mm) were randomly divided into two groups (n = 25). Mice in EC group were i.t. injected with E. coli (1×10^5 , 50 μ L in PBS). The length and width of the mice's tumor were measured every two days after injection. Four mice from each group were randomly chosen and sacrificed at each time point (i.e., 0, 0.5, 3, 7 and 15 days after injection). The tumors were collected and three in each group were weighted. Under a sterile tactical environment, each tumor was cut into three portions (~100 mm³ each) for IHC, cytokine profile, and bacteria burden analysis. The tumor slices were deparaffinized, rehydrated, and then incubated with primary anti-CD3 (Ablonal), anti-CD86 (Ablonal), anti-ARG-1 (Ablonal), and secondary antibodies, followed by staining with DAB and hematoxylin. Fluorescent microscopy (Leica) was used to visualize the slides, and the data were analyzed using Image J."

x) Justification of co-culturing 4T1 and BMDM to monitor HMGB release is not described. Will 4T1 still produce HMGB in the absence of BMDM?

Answer: We thank the reviewer for raising this comment. Co-culturing the intended cancer cells with BMDM is a commonly-used approach to monitor HMGB1 release upon the phagocytic clearance of macrophages [*Nat. Biomed. Eng.* **4**, 1102–1116 (2020)]. Following this reviewer's suggestion, we have carried out additional experiments to evaluate the HMGB1 release in the present/absence of BMDM *in vitro*. As shown in Fig. R1, the HMGB1 level in the supernatants of 4T1 cells after being treated with Au@Ag₂Se-FA plus NIR irradiation (10 min) was 382% as that of Control. As PI-Au@Ag₂Se-FA can increase the ROS generation in tumor cells (Supplementary Fig. 15) to further promote the HMGB1 release from dying 4T1 cells [*Nat. Rev. Cancer* **12**, 860–875 (2012), *Angew.*

Chem. Int. Ed. **60**, 4657–4665 (2021)], the average HMGB1 release for 4T1 cells treated by NPs+NIR then co-cultured with BMDMs for 12 h (NPs+NIR) was 121% as that of C-NPs+NIR. We have included the results in Fig. 4d and described them together with the justification of co-culturing 4T1 and BMDM in the revised manuscript (Page 7) as follows:

Fig. R1 | HMGB1 level in the supernatants of 4T1 cells after various treatments, including i) 4T1 cells co-cultured with BMDMs for 12h (Control), ii) 4T1 cells culture with Au@Ag₂Se-FA for 12 h then co-cultured with BMDMs for another 12h (Au@Ag₂Se-FA), iii) 4T1 cells treated by NPs+NIR for 10 min without further incubation (C-NPs+NIR), and iv) 4T1 cells treated by NPs+NIR and co-cultured with BMDMs for 12h (NPs+NIR).

“Next, we cultured 4T1 cells with bone marrow-derived macrophages (BMDMs) (Fig. 4d) to evaluate the release of high mobility group box1 (HMGB1) upon macrophage phagocytic clearance [22]. Au@Ag₂Se-FA alone could not regulate HMGB1. The high HMGB1 level of NPs+NIR (464% as that of Control) indicated secondary necrosis and enhanced immune responses [37,38] even upon macrophage clearance.”

Fig. 4 | Therapeutic efficacy and immune response induced by Au@Ag₂Se-FA in vitro. **d**, HMGB1 level in the supernatants of 4T1 cells after various treatments, including i) 4T1 cells co-cultured with BMDMs for 12 h (Control), ii) 4T1 cells culture with Au@Ag₂Se-FA for 12 h then co-cultured with BMDMs for another 12 h (Au@Ag₂Se-FA), iii) 4T1 cells treated by NPs+NIR and co-cultured with BMDMs for 12 h (NPs+NIR).

y) Certain other elements are missing in the Method section. Please provide more info throughout such that the work can be reproduced in the future.

Answer: We are grateful to the reviewer for the valuable advice. Following this reviewer’s suggestions, we have

added the details in the 'Methods' section of the revised manuscript accordingly.

- For example, sample preparation method for SEM imaging is not provided. It is unclear what diluted RBCs dispersion means (and how it's prepared).

"The morphology of red blood cells (RBCs) and E. coli cells were imaged by scanning electron microscopy (SEM, JME-7500F). The NPs dispersions (0.9 mL) at various concentrations (20–1000 $\mu\text{g}\cdot\text{mL}^{-1}$) were added in a centrifuge tube (1.5 mL) followed by addition of diluted RBCs dispersion (1mL RBCs diluted in 10 mL PBS, 0.1 mL) and incubated at 37 °C for 24 h. The diluted RBCs dispersion after incubation with the NPs was then applied for SEM imaging."

- Cell number for 4T1 in in vitro therapeutic effect is missing.

"4T1 cells (1.0×10^5 cells per well) were seeded in 96-well plate overnight, then incubated with Au@Ag₂Se NPs dispersion ($100 \mu\text{g}\cdot\text{mL}^{-1}$)."

- Mouse model and treatment deserves its own section. Currently, they are mentioned across two sections in the method. The readers need to combine info across two sections to understand the experiment. Timeline for all in vivo studies need clarification (e.g., how many days after implantation did the treatment start? mouse age? mouse vendor?).

Following this reviewer's suggestion, we list the mice model and treatment as a separate section in 'Methods' and added the details as follows:

***"Establishment of mice tumor model.** 1.0×10^6 4T1 cells (or RM-1 cells) (50 μL in PBS) were subcutaneously injected into the right back of four-week-old female (male for RM-1 model) BALB/c mice (Charles River, Beijing). To construct the dual-tumor model, 1.0×10^6 4T1 cells were subcutaneously injected into the right back of each mouse to establish the primary tumor, and 5.0×10^5 4T1 cells were subcutaneously injected into the left back of each mouse to obtain the distal tumor. When the tumors grew to ~8 mm in diameter, the tumor-bearing mice were randomly divided into different groups for various treatments. All animal experiments were carried out under protocols approved by the Institutional Animal Care and Use Committee. The approval number of animals is IACUC-2019021."*

- For in vivo efficacy, was the endpoint set at 16 days post-treatment? were separate studies with another set of mice conducted for survival analysis? Or were they conducted at the same time and a subset of mice were sac'ed at 16 days?

"After the observation at Day 16, five mice from each group were kept for survival analysis, and another nine mice from each group (T5 and T6) were re-challenged with 4T1 tumor cells by subcutaneous administration (R-T5, R-T6). The survival analysis was repeated three times using three separate mice groups (n = 5). Except those for survival analysis, the rest mice were sacrificed after all the experiments, following the Ethics Committee for Animal Experimentation guidelines."

5. Issues on rigor and Statistics

z) Statistic analysis is not performed for tumor growth curve and survival. These need different analyses beyond one-way ANOVA.

Answer: We thank the reviewer for raising this advice. Following this reviewer's suggestion, we have carried out statistical analysis for tumor growth curve and survival. For the significance between two groups, a two-tailed Student's *t*-test was employed. For multiple comparisons, one-way analysis of variance (ANOVA) with Tukey's post hoc test was used.

REVIEWERS' COMMENTS

Reviewer #1 (Remarks to the Author):

I have read the revised manuscript, reviewer comments, and the authors' responses with great interest. The authors have provided a scientific response and plenty of experimental data to fully address my concerns, which greatly supports the intention of this manuscript and improves the importance of the manuscript. The revised version of the manuscript has improved substantially. Thus, I recommend the acceptance in the journal as it is.

Reviewer #3 (Remarks to the Author):

The authors addressed all my comments/questions satisfactorily except that the requested clone number/information of all antibodies used for both flow cytometry and IHC are still not provided (e.g., mouse-CD3 (clone 145-2C11), etc.). The authors responded with only vendor information. Once these info are provided, the reviewer would recommend for publication. Also, there are still small grammatical errors throughout, and thus another round of thorough proofreading should be performed by either the authors or the publisher.

Reviewer #4 (Remarks to the Author):

In this revised version, the authors have addressed most concerns raised by the reviewers. However, this reviewer has some further suggestions and comments that the authors may be considered to further improve the current manuscript.

Personally, I don't like "Sterilization" and think it is not the appropriate word in the settings of the study. Sterilization is defined as a process of complete elimination or destruction of all forms of microbial life (i.e., both vegetative and spore forms), which is carried out by various physical and chemical methods. Technically, there is reduction $\geq 10^6$ log colony forming units (CFU) of the most resistant spores achieved at the half-time of a regular cycle. While in the current study, no evidence for complete removal of all microbial life. Furthermore, it is the term for hospital or Medical Devices and Equipment. In the current setting, it is more related to drugs or antitumor agents.

The authors mentioned that Fenton-like activities of Ag. Please be aware that it is specially referring to transition metals, such as iron or copper. Ag should not be as active as those metals. Please be cautious to use the term. There should be a cycle between Fe²⁺-3 or Cu¹⁺-2. Is there a such electron transfer process?

Please discuss the feasibility for radiation assisted treatment option for future clinical settings.

For the so-called non-sterile conditions, please discuss it in a broad aspect for the comparison the current study with other related studies.

For the generality study, there are no two models investigated, it is recommended that don't use this broad term to describe.

For the layout and design of Figure 1, it is too cheesy and out of focus.

Comments:

Reviewer #1

I have read the revised manuscript, reviewer comments, and the authors' responses with great interest. The authors have provided a scientific response and plenty of experimental data to fully address my concerns, which greatly supports the intention of this manuscript and improves the importance of the manuscript. The revised version of the manuscript has improved substantially. Thus, I recommend the acceptance in the journal as it is.

Answer: We highly appreciate the reviewer's time and effort in re-evaluate our manuscript and we are glad that the reviewer recommends accepting our work in Nature Communications as it is.

Comments:

Reviewer #3

The authors addressed all my comments/questions satisfactorily except that the requested clone number/information of all antibodies used for both flow cytometry and IHC are still not provided (e.g., mouse-CD3 (clone 145-2C11), etc.). The authors responded with only vendor information. Once these info are provided, the reviewer would recommend for publication. Also, there are still small grammatical errors throughout, and thus another round of thorough proofreading should be performed by either the authors or the publisher.

Answer: We highly appreciate the reviewer's time and effort in re-evaluate our manuscript and making the recommendation for publication in this journal. Following this reviewer's suggestions, we have included the clone number/information of all antibodies used for flow cytometry and IHC in the revised manuscript (Pages 16 and 17) as follows:

"The tumor slices were deparaffinized, rehydrated, and then incubated with primary anti-CD3 (Ablonal, A1753, dilution 1:100), anti-CD86 (Ablonal, A2352, dilution 1:100), anti-ARG-1 (Ablonal, A4923, dilution 1:100), and secondary antibodies (HRP Goat anti-rabbit IgG, Ablonal, AS014, dilution 1:100), followed by staining with DAB and hematoxylin."

"The anti-mouse of CD3-PE (clone: 17A2), CD4-APC-cy7 (clone: GK1.5), CD8a-FITC (clone: 53-6.7), CD11b-PE (clone: M1/70), F4/80-APC (clone: BM8), CD86-PE-Cy7 (clone: GL-1), CD206-FITC (clone: C068C2), CD49b-APC (clone: HMA2), TCRβ-FITC (clone: H57-597), CD44-PE (clone: IM7), and CD62L-APC-Cy7 (clone: MEL-14) were purchased from Biolegend."

Moreover, we double-checked the manuscript for grammatical errors as suggested by the reviewer.

Comments:

Reviewer #4

In this revised version, the authors have addressed most concerns raised by the reviewers. However, this reviewer has some further suggestions and comments that the authors may be considered to further improve the current manuscript.

Answer: We are grateful to the reviewer for taking the time to evaluate our revision and providing constructive comments. Following this reviewer's suggestions, we have made changes and added discussion accordingly in the revised manuscript. Please refer to our point-by-point responses below.

--- Personally, I don't like "Sterilization" and think it is not the appropriate word in the settings of the study. Sterilization is defined as a process of complete elimination or destruction of all forms of microbial life (i.e., both vegetative and spore forms), which is carried out by various physical and chemical methods. Technically, there is reduction $\geq 10^6$ log colony forming units (CFU) of the most resistant spores achieved at the half-time of a regular cycle. While in the current study, no evidence for complete removal of all microbial life. Furthermore, it is the term for hospital or Medical Devices and Equipment. In the current setting, it is more related to drugs or antitumor agents.

Answer: We appreciate the reviewer's suggestion and have replaced "Sterilization" with "*Antibacteria*" or "*Bacteria killing*" in the revised manuscript.

--- The authors mentioned that Fenton-like activities of Ag. Please be aware that it is specially referring to transition metals, such as iron or copper. Ag should not be as active as those metals. Please be cautious to use the term. There should be a cycle between Fe^{2-3} or Cu^{1-2} . Is there a such electron transfer process?

Answer: The cycle between Fe^{2+} and Fe^{3+} is known as the Fenton reaction, whereas the Fenton-like reactions refer to a type of reaction in which other transition metals (such as Co, Cd, Cu, Ag, Mn, Ni, etc.) and other iron-containing compounds catalyze H_2O_2 . It has been well demonstrated that Ag can undergo a Fenton-like reaction in the presence of H_2O_2 [*Chem. Eng. J.* 466, 143147 (2023); *Small* 18, 2103868 (2022)]. In the Ag-participated Fenton-like process, the redox cycle of Ag^+/Ag^0 initiates $\cdot OH$ generation from H_2O_2 [e.g., *Biomaterials* 33, 7547–7555 (2012); *Environ. Chem. Lett.* 19, 2405–2424 (2021)]. Considering the weak acidity and excess H_2O_2 in tumors, the Fenton-like reaction of Ag can be used for chemodynamic therapy [e.g., *Chem. Eng. J.* 400, 125949 (2020)].

--- Please discuss the feasibility for radiation assisted treatment option for future clinical settings.

Answer: We highly appreciate the reviewer's valuable advice. Following this reviewer's suggestion, we have included discussion regarding the potential of the present agent for radiation treatment in the revised manuscript (Page 12) as follows:

“Furthermore, considering the high-Z and high X-ray attenuation coefficient of Au [33] and the X-ray absorption-induced photoelectric or Auger electron generation by Ag NCs [49], Au@Ag₂Se-FA could also be a potential candidate for radiation treatment and computed tomography imaging.”

49. Tsang, G. Y. & Zhang, Y. Nanomaterials for light-mediated therapeutics in deep tissue. *Chem. Soc. Rev.* (2024). DOI: 10.1039/d3cs00862b.

--- For the so-called non-sterile conditions, please discuss it in a broad aspect for the comparison the current study with other related studies.

Answer: We highly appreciate the reviewer’s constructive comments. Following the reviewer’s suggestion, we have included more description for the non-sterile conditions in the revised manuscript (Page 10) as follows:

“Compared to direct bacterial injection, cultivating mice in a non-sterile environment can better imitate real-life conditions that are more relevant to clinical scenarios. External bacteria have been demonstrated in clinics and laboratories to infect the host, causing accumulation and residence in tumors [5,6].”

--- For the generality study, there are no two models investigated, it is recommended that don’t use this broad term to describe.

Answer: Following this reviewer’s suggestion, we have modified the title for the prostate model therapy section to be **“Therapeutic efficacy on prostate cancer model.”** in the revised manuscript (Page 11).

--- For the layout and design of Figure 1, it is too cheesy and out of focus.

Answer: Figure 1 depicts the major content of this work, including the design, multiple anti-tumor modes, antibacterial activities, and modulation of tumor microenvironment immunity of our nanoagent. Following the criteria of the Nature journals, we intend to present readers with the primary information and conclusions of this article directly in Figure 1 to improve readability and time efficiency. We regret that the reviewer thought this figure was *“too cheesy and out of focus”*. Obviously, we hold different views on aesthetics and layout.